# Stop Training for the Worst:
# Progressive Unmasking Accelerates Masked Diffusion Training

Jaeyeon Kim [* 1]   Jonathan Geuter [* 1 2]   David Alvarez-Melis [1 2]   Sham Kakade [1 2]   Sitan Chen [1]

## Abstract

Masked Diffusion Models (MDMs) have emerged as a promising approach for generative modeling in discrete spaces. By generating sequences in any order and allowing for parallel decoding, they enable fast inference and strong performance on non-causal tasks. However, this flexibility comes with a *training complexity* trade-off: MDMs train on an exponentially large set of masking patterns, which is not only computationally expensive, but also creates a train–test mismatch between the random masks used in training and the highly structured masks induced by inference-time unmasking. In this work, we propose Progressive UnMAsking (PUMA), a simple modification of the forward masking process that aligns training-time and inference-time masking patterns, thereby focusing optimization on *inference-aligned masks* and speeding up training. Empirically, PUMA speeds up pretraining at the 125M scale by $\approx$ $2.3\times$ and offers complementary advantages on top of common recipes like autoregressive initialization. We open-source our codebase at https://github.com/JaeyeonKim01/PUMA.

## 1. Introduction

In recent years, Masked Diffusion Models (MDMs) (Lou et al., 2023; Shi et al., 2024; Sahoo et al., 2024) have emerged as a compelling alternative to autoregressive models for discrete generative modeling, driven by rapid scaling efforts from 7B (Ye et al., 2025; Nie et al., 2025; Gong et al., 2025; Song et al., 2025) to 100B parameters (Bie et al., 2025), as well as industry-scale deployments (Labs et al., 2025; DeepMind, 2025). Much of their appeal comes from inference-time advantages: parallel decoding speeds

[*]Equal contribution [1]Harvard University [2]Kempner Institute. Correspondence to: Jaeyeon Kim <jaeyeon_kim@g.harvard.edu>.

*Proceedings of the 43rd International Conference on Machine Learning*, Seoul, South Korea. PMLR 306, 2026. Copyright 2026 by the author(s).

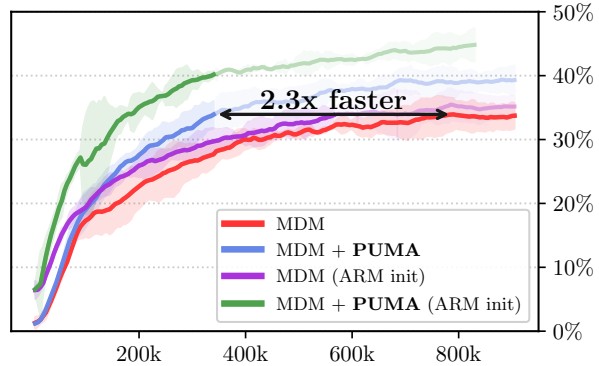

*Figure 1.* **PUMA** (blue) accelerates Masked Diffusion Model training (red) by changing the forward process. Experiment on 125M-scale, from scratch trained on TinyGSM (Liu et al., 2023). Moreover, PUMA is compatible with autoregressive initialization (purple curve over green curve). Error bars are 95% CIs.

up generation, while any-order inference better accommodates tasks that require non-autoregressive generation.

At the same time, MDMs come with a well-known trade-off (Kim et al., 2025b), **Train for the Worst**: to have any-order generation ability, they must train on unmasking tasks with an exponential number of masking patterns, substantially increasing the complexity of training. Under this paradigm, compute is spread across all masking patterns, while at inference, the masking patterns occupy a negligible fraction of the ones seen during training. This is because adaptive inference prioritizes positions for which the model's predictions are most certain, causing unmasking decisions to concentrate on a small subset of masking patterns.

*Yet*, this mismatch can be turned into an opportunity: training complexity can be reduced by focusing on inference-aligned masking patterns. However, despite training efficiency being essential to scaling MDMs, recent work has largely emphasized inference, leaving **MDM training comparatively underexplored**.

In this work, we propose **Progressive UnMAsking (PUMA)**. With a *modified forward process*, PUMA generates training examples whose masking patterns *provably match* those observed at inference, thereby resolving the train-test mismatch. To our knowledge, PUMA is the first

approach to **accelerate discrete diffusion training** by re-designing the forward masking process itself, rather than architectural (Arriola et al., 2025b; Ni & team, 2025) or recipe-level remedies, such as autoregressive initialization (Bie et al., 2025; Ye et al., 2025; Gong et al., 2025; DeepMind, 2025) or block-size curriculum (Bie et al., 2025; Fan et al., 2026), or loss reweighting (Shi & Titsias, 2025).

Empirically, PUMA accelerates MDM pretraining by up to $2.3\times$ in iteration–accuracy comparisons at the 125M scale, while introducing only *one* additional hyperparameter and incurring no forward-pass overhead. We further demonstrate that PUMA remains effective in more realistic training scenarios, including autoregressive initialization, where we observe $4.0\times$ speedup, and post-training of a 7B-scale pretrained MDM.

## 2. Preliminaries

We begin by reviewing the training and inference of Masked Diffusion Models (MDMs).

**Notation.** Assume we aim to learn the distribution $p_{\text{data}}$ over sequences of length $L$ with finite vocabulary $\mathcal{V}$. Denote by $\mathbf{x}^i$ the $i$-th element of a given sequence $\mathbf{x} = (\mathbf{x}^1, \ldots, \mathbf{x}^L)$, and let $\Delta(\mathcal{V})$ be the simplex of probability distributions over $\mathcal{V}$.

**MDM training.** MDMs (Lou et al., 2023; Shi et al., 2024; Sahoo et al., 2024) introduce an auxiliary mask token $\mathbf{m}$. At a high level, they learn the posterior marginal distributions over $\mathcal{V}$ at each mask token, conditioned on a partially masked sequence. Concretely, an MDM defines a *forward masking process*: for a given $\mathbf{x}_0 \sim p_{\text{data}}$, draw a time step $t \sim \text{Unif}[0, 1]$, then independently replace each token $\mathbf{x}_0^i$ by $\mathbf{m}$ with probability $t$.[1] This results in a partially masked sequence $\mathbf{x}_t \in (\mathcal{V} \cup \{\mathbf{m}\})^L$.

This resulting joint distribution over $(\mathbf{x}_0, t, \mathbf{x}_t)$ induces the coordinate-wise posterior $p(\mathbf{x}_0^i = \cdot \mid \mathbf{x}_t, t)$, which we refer to as *unmasking posterior*: the distribution over clean token at masked position $i$ given $(\mathbf{x}_t, t)$. The masking process above has an intriguing property–the unmasking posterior is *time-agnostic* (Ou et al., 2024; Zheng et al., 2024); it depends only on the observed (partially masked) sequence $\mathbf{x}_t$, not the time step $t$; therefore, we drop the time step in the notation and write $p(\mathbf{x}_0^i = \cdot \mid \mathbf{x}_t) = p(\mathbf{x}_0^i = \cdot \mid \mathbf{z})$, adopting the notation $\mathbf{z} = \mathbf{x}_t$.

To learn this unmasking posterior, MDMs employ a neural network $f_\theta$ that takes a sequence $\mathbf{z}$ as an input and outputs a tensor of shape $|\mathcal{V}| \times L$. Its $i$-th column $f_\theta^i(\cdot \mid \mathbf{z}) \in \Delta(\mathcal{V})$ learns the unmasking posterior $f_\theta^i(v \mid \mathbf{z}) \approx p(\mathbf{x}_0^i = v \mid \mathbf{z})$. To train $f_\theta$, MDMs minimize the following cross-entropy

---

[1]We present the masking process using a linear schedule.

loss summed over all masked indices:

$$\mathcal{L}(\theta) = \mathbb{E}_{(\mathbf{x}_0, t, \mathbf{x}_t)} \left[ \frac{1}{t} \sum_{i:\, \mathbf{x}_t^i = \mathbf{m}} -\log f_\theta^i(\mathbf{x}_0^i \mid \mathbf{x}_t) \right].$$

where the expectation is taken over the joint distribution of $(\mathbf{x}_0, t, \mathbf{x}_t)$ defined above. As desired, the unique minimizer $f_{\theta^\star}$ of the loss $\mathcal{L}(\theta)$ satisfies $f_{\theta^\star}^i(v \mid \mathbf{z}) = p(\mathbf{x}_0^i = v \mid \mathbf{z})$. This viewpoint connects the MDM training to any-order masked language models such as BERT (Devlin et al., 2019), which also model the posterior marginals at masked positions.

**MDM inference.** MDM inference starts from a fully masked length-$L$ sequence $\mathbf{x}_1 = (\mathbf{m}, \ldots, \mathbf{m})$ and proceeds over a monotonically decreasing grid of times $t_0 = 1 > \cdots > t_l = 0$. At each step $t_j$, given a partially masked sequence $\mathbf{x}_{t_j} \in (\mathcal{V} \cup \{\mathbf{m}\})^L$, inference proceeds by two steps to obtain $\mathbf{x}_{t_{j+1}}$:

(a) Choose a subset of masked positions $\mathcal{S}$ to unmask.

(b) for all $i \in \mathcal{S}$, unmask $\mathbf{x}_{t_j}^i$ with a clean token from $f_\theta^i(\cdot \mid \mathbf{x}_{t_j}) \in \Delta(\mathcal{V})$.

The flexibility of choosing $\mathcal{S} \subseteq \{i \mid \mathbf{x}_{t_j}^i = \mathbf{m}\}$ lies at the core of MDM inference; selecting which positions to unmask next can dramatically influence downstream performance both in terms of accuracy, as well as efficiency. A large body of prior work (Chang et al., 2022; Zheng et al., 2023; Kim et al., 2025b; Nie et al., 2025; Peng et al., 2025a; Jazbec et al., 2025; Hong et al., 2025; Ma et al., 2025; Lee et al., 2025; Hayakawa et al., 2025; Mo et al., 2025) investigates this question.

Formally, an unmasking policy $g_\phi$ – either heuristic or learned – maps a partially masked sequence and time $(\mathbf{x}_t \in (\mathcal{V} \cup \{\mathbf{m}\})^L, t)$ to a (possibly stochastic) subset $\mathcal{S} \subseteq \{i : \mathbf{x}_t^i = \mathbf{m}\}$ of the masked positions, which is used at each inference step to select a subset to unmask. A widely used instantiation scores each masked position and then sets $\mathcal{S}$ as the top-K positions: $g_\phi(\mathbf{x}_t, t) = \text{TopK}_{i:\, \mathbf{x}_t^i = \mathbf{m}} [\text{score}(i)]$, where $\text{score}(i)$ is a confidence score for position $i$. Common choices for computing the score include the maximum predicted probability $\max_{v \in \mathcal{V}} f_\theta(v \mid \mathbf{x}_t, t)$ ("Top-K"), the margin between the top two probabilities $f_\theta(v_1 \mid \mathbf{x}_t, t) - f_\theta(v_2 \mid \mathbf{x}_t, t)$ where $v_1, v_2$ are the top-2 tokens with the highest assigned probability ("Top-K margin"), the negative entropy of the categorical distribution ("entropy-based"), or a position-dependent schedule (e.g., semi-autoregressive inference).

### 2.1. Prior work on accelerating MDM training

Given the any-order nature of the prediction task, MDM training is inherently more challenging and less compute-efficient than autoregressive training (Nie et al., 2024; Kim

et al., 2025b), where the model only needs to learn causal next-token prediction. Consequently, prior work has relied on architectural or training-recipe-level strategies applied on top of the core MDM procedure. At scale, a common practice is to initialize MDMs from pretrained autoregressive models (Ye et al., 2025; Song et al., 2025; Bie et al., 2025; Fan et al., 2026), along with mixture-of-experts variants (Ni & team, 2025). Block diffusion (Arriola et al., 2025a;b) combines block-wise autoregressive generation with MDM-style inference within each block, yielding speedups but essentially *trading off* full any-order flexibility: at each step, the set of unmasked positions must lie within a block. **Notably, these approaches typically leave the underlying masking process unchanged.** In contrast, PUMA operates **at the level of the forward masking process**; it is thus largely orthogonal to these designs and can be incorporated alongside them, potentially compounding speedups.

PAPL (Peng et al., 2025b), a recently proposed MDM training framework, also studies the train–test mismatch of MDMs. While the authors derive an unmasking-policy-aware ELBO which is closely related to PUMA's setting, their method retains the standard random masking process at training and instead only applies loss reweighting across masked positions, which leaves the distributional train–test mismatch unchanged. We defer a more detailed comparison between PAPL and PUMA to Section 5.[2]

## 3. PUMA: Progressive UnMAsking

In this section, we introduce **Progressive UnMAsking** (**PUMA**). In Section 3.1, we establish the theoretical foundation of PUMA and present how it resolves the MDM's train-test mismatch stated in the introduction. We then describe our empirical design choices for PUMA in Section 3.2. In Section 3.3, we instantiate examples of distributions for which aligning training-time masking with inference-time structure provably improves the statistical efficiency.

### 3.1. Theoretical foundation of PUMA

We begin by formalizing the train–test mismatch in MDMs.

**Train-test mismatch of MDM.** Recall that MDM inference proceeds by following an unmasking policy $g_\phi$ that looks at the *currently revealed* tokens and decides which masked positions to unmask next. As a consequence, inference does not visit masked sequences uniformly (as done during training); instead, it induces a policy-dependent distribution over intermediate masked sequences. For example, under a *semi-autoregressive* unmasking policy (Nie et al., 2025),

inference proceeds block-wise in an autoregressive manner, hence the maskings observed at inference are restricted to block-wise causal patterns.

Meanwhile, training allocates compute uniformly across all masking patterns, since each position is masked i.i.d. This suggests that training could be more compute-efficient if we *concentrate* the training distribution on masking patterns that arise at inference. This raises the question:

*Is there a masking procedure that resolves this mismatch?*

**Challenge.** The above question is subtle for two reasons. **(1)** At training time, we start from a clean sequence $\mathbf{x}_0$ and construct a masked example $\mathbf{x}_t$; at inference, however, intermediate states $\mathbf{x}_t$ are generated by *progressively* applying the unmasking policy to partially masked sequences, without access to any clean sequence. **(2)** We cannot arbitrarily change the forward process: modifying the joint distribution over $(\mathbf{x}_0, t, \mathbf{x}_t)$ can inadvertently change what MDM training aims to learn. Concretely, unless done carefully, changing $(\mathbf{x}_0, t, \mathbf{x}_t)$ can shift the minimizer of the loss

$$\mathcal{L}(\theta) = \mathbb{E}_{(\mathbf{x}_0, t, \mathbf{x}_t)} \left[ \frac{1}{t} \sum_{i \,:\, \mathbf{x}_t^i = \mathbf{m}} - \log f_\theta^i(\mathbf{x}_0^i \mid \mathbf{x}_t) \right]. \quad (1)$$

**PUMA.** We now present PUMA and explain how it resolves these challenges. The *crux* of PUMA is the so-called *teacher-forced chain*, which we use to generate training-time samples.

At a high level, the teacher-forced chain is a progressive unmasking process that follows the *same unmasking policy* $g_\phi$ used at inference time, but instead of sampling clean tokens from a posterior, it reveals the corresponding ground-truth tokens from the data sample $\mathbf{x}_0$. Surprisingly, this simple chain lets us align the distribution over masking patterns at training-time with the one at inference time.

**Teacher-forced chain.** Fix a clean sequence $\mathbf{x}_0$, an unmasking policy $g_\phi$, and an integer $l \geq 1$. Define a time grid $t_j := 1 - j/l$ for $j = 0, 1, \ldots, K$, so that $t_0 = 1$ and $t_l = 0$. We initialize the chain at the fully masked state $\mathbf{x}_{t_0} = \mathbf{x}_1 = (\mathbf{m}, \ldots, \mathbf{m})$. To clarify, in contrast to inference (Section 2), we do not draw a clean token from $f_\theta$; whenever positions are selected, we reveal ground-truth tokens from $\mathbf{x}_0$. Concretely, given a partially masked state $\mathbf{x}_{t_j}$, obtain $\mathbf{x}_{t_{j+1}}$ by:

(a) Choose a subset of masked positions $\mathcal{S} = g_\phi(\mathbf{x}_{t_j}, t_j)$.

(b) For each $i \in \mathcal{S}$, unmask $\mathbf{x}_{t_j}^i = \mathbf{m}$ into clean token $\mathbf{x}_0^i$.

The chain above is defined *conditional* on a particular $\mathbf{x}_0$, i.e., it specifies a conditional distribution over trajectories $(\mathbf{x}_{t_0}, \ldots, \mathbf{x}_{t_l})$ given $\mathbf{x}_0$. At training time, we first sample $\mathbf{x}_0 \sim p_{\text{data}}$ and then run the chain (conditioned on $\mathbf{x}_0$).

---

[2]The authors realized the theoretical connection with (Peng et al., 2025b) after the release of the first version. We therefore include a thorough comparison in the second version of our manuscript.

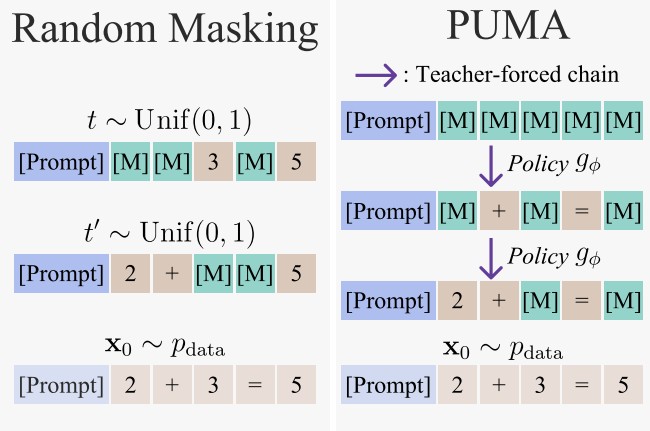

*Figure 2.* **Illustration of PUMA.** Under PUMA, training examples are generated via the teacher-forced chain using a given $\mathbf{x}_0$ (purple), the current model's policy $g_\phi$, and clean tokens $\mathbf{x}_0$. In contrast, standard MDM training yields (independently drawn) training samples by random masking.

Therefore, each chain run yields multiple intermediate masked sequences, each of which serves as a training example paired with the ground-truth sequence $\mathbf{x}_0$. In contrast, standard MDM training generates a single training example with random masking. We illustrate this in Figure 2. This teacher-forced chain can be efficiently implemented in practice without any computational overhead, and we will revisit it in Section 3.2.

We now informally state the **theoretical guarantee of PUMA**: *marginal agreement* and *minimizer preservation*.

> **(PUMA in a nutshell).** PUMA training, i.e., sampling $\mathbf{x}_0 \sim p_{\text{data}}$ and generating contexts $\mathbf{x}_{t_j}$ via the teacher-forced chain, **(i)** matches the marginal distributions induced by MDM inference under the same policy $g_\phi$ and **(ii)** preserves the unmasking-posterior minimizer of the MDM objective.

Together, these two properties resolve the challenges identified earlier: PUMA aligns training with the inference-time distribution induced by an unmasking policy, while ensuring that modifying the forward process does not change what the model is trained to learn. *As a result, PUMA enables more compute-efficient MDM training without affecting the minimizer of the underlying objective.*

**Current model as an unmasking policy.** An important practical consideration in operating the teacher-forced chain during pretraining is that the final unmasking policy, namely the one used at inference, is not available a priori: it is induced by the final pretrained network (see Section 2 for the policy instances). Accordingly, we construct the teacher-forced chain using the unmasking policy defined by the *current model* at that point in the training process.

Intuitively, as training proceeds and the model's predictions get better, the induced policy $g_\phi$ rapidly approaches the inference-time policy of the final model. In particular, the *ranking* of which positions to unmask might tend to stabilize earlier than full posterior calibration, so the policy becomes *approximately final* relatively early in training. We indeed validate this empirically in Section 4.1, and return to implementation details in Section 3.2.

### 3.1.1. PUMA'S MARGINAL AGREEMENT PROPERTY

In this section, we formalize the marginal agreement property of PUMA. The proof is deferred to Appendix A.

**Intuition.** The unmasking posterior $p(\mathbf{x}_0^i = \cdot \mid \mathbf{z})$ used in the MDM inference step (b) is the data conditional distribution given $\mathbf{z}$. Therefore, "revealing a token from the *true* posterior" (*idealized* MDM inference) is distributionally equivalent to "first draw $\mathbf{x}_0 \sim p_{\text{data}}$, then reveal its tokens, and later marginalize over $\mathbf{x}_0 \sim p_{\text{data}}$" (PUMA): both generate the same joint law over revealed tokens and remaining masks. Since $g_\phi$ only reacts to what is currently revealed (and time), both procedures induce the same distribution over intermediate masked contexts at each $t$.

**Proposition 1** (Informal). *Fix a policy $g_\phi$ and consider an idealized MDM inference procedure that, at step (b), samples clean tokens from the ground-truth unmasking posterior $p(\mathbf{x}_0^i = \cdot \mid \mathbf{z})$. Let $q_t$ denote the distribution of $\mathbf{x}_t$ under this idealized inference. Then, for every $t$, the marginal distribution of $\mathbf{x}_t$ induced by the teacher-forced chain with $\mathbf{x}_0 \sim p_{\text{data}}$ coincides with $q_t$.*

**Remark.** The precise way to present Proposition 1 is under the *idealized* policy, where we only choose one position to unmask at each time step. This is because the token sampling step at inference is always done with $f_\theta$'s per-position posterior; the distributional dependency across positions is ignored. A more nuanced way to understand the teacher-forced chain is through continuous-time Markov chains (Campbell et al., 2022; Gat et al., 2024; Kim et al., 2025a), under which the marginal agreement holds for all $t \in [0, 1]$. We formalize this argument in Appendix A.1.

### 3.1.2. PUMA'S MINIMIZER GUARANTEE

In this section, we demonstrate that PUMA does *not* alter the minimizer of the training loss despite intervening in the distribution $(\mathbf{x}_0, t, \mathbf{x}_t)$.

**Intuition.** We first investigate how the design of the forward process can influence the minimizer of the training loss. Consider the i.i.d. forward process, whose conditional distribution takes the following form:

$$p(\mathbf{x}_t = \mathbf{z} \mid \mathbf{x}_0, t)$$
$$= (1 - t)^{|\mathbf{um}(\mathbf{z})|} t^{L - |\mathbf{um}(\mathbf{z})|} \cdot \mathbf{1}\{\mathbf{x}_0^{\mathbf{um}(\mathbf{z})} = \mathbf{z}^{\mathbf{um}(\mathbf{z})}\},$$

---

**Algorithm 1** PUMA pseudocode (one training iteration)

---

1: *Require:* MDM $f_\theta$, per-stage unmask number $K \in \mathbb{N}$, sequence length $L$, masked sequences $\{\mathbf{z}(j)\}_{j=1}^B$, paired clean samples $\{\mathbf{x}_0(j)\}_{j=1}^B$, and per-example stage counters $\{n(j)\}_{j=1}^B \in [0, L/K]$

2: **Forward pass:** compute logits $\{f_\theta(\cdot \mid \mathbf{z}(j))\}_{j=1}^B$                $\triangleright B \times L \times \Delta(\mathcal{V})$

3: Compute loss $\mathcal{L}(\theta)$ from logits and $\mathbf{x}_0(j)$; update $\theta$ with the optimizer                $\triangleright$ training update

4: # PUMA streaming update (per example j), parallelized

5: **if** $n(j) = L/K$ **then**

6:     Sample $\mathbf{x}_0(j) \sim p_{\text{data}}$; set $n(j) \leftarrow 0$; initialize $\mathbf{z}(j)$                $\triangleright$ refill a new sequence, restart the chain

7: **else**

8:     Compute scores $s_i \leftarrow \max_{v \in \mathcal{V}} f_\theta^i(v \mid \mathbf{z}(j))$ for masked indices $i$                $\triangleright$ instantiate unmasking policy from the model

9:     Select reveal set $\mathcal{S}(j)$ using $\{s_i\}$                $\triangleright$ top-K + thresholding

10:     For all $i \in \mathcal{S}(j)$, update $\mathbf{z}(j)^i \leftarrow \mathbf{x}_0(j)^i$, update $n(j)$                $\triangleright$ teacher-forced chain update

---

where $\mathbf{um}(\mathbf{z}) := \{i \mid \mathbf{z}^i \neq \mathbf{m}\}$ is the set of unmasked tokens $\mathbf{z}$ and $\mathbf{1}\{\mathbf{x}_0^{\mathbf{um}(\mathbf{z})} = \mathbf{z}^{\mathbf{um}(\mathbf{z})}\}$ is the indicator function that identifies whether $\mathbf{z}$ matches $\mathbf{x}_0$ on $\mathbf{um}(\mathbf{z})$. Now we observe how the first term (green factor) cancels out in the Bayes-optimal predictor, which is also the cross-entropy training loss's minimizer. By Bayes' rule,

$$p(\mathbf{x}_0 \mid \mathbf{x}_t = \mathbf{z}, t) \propto p(\mathbf{x}_t = \mathbf{z} \mid \mathbf{x}_0, t)\, p_{\text{data}}(\mathbf{x}_0)$$

$$= (1-t)^{|\mathbf{um}(\mathbf{z})|}\, t^{L-|\mathbf{um}(\mathbf{z})|} \cdot \mathbf{1}\{\mathbf{x}_0^{\mathbf{um}(\mathbf{z})} = \mathbf{z}^{\mathbf{um}(\mathbf{z})}\}\, p_{\text{data}}(\mathbf{x}_0),$$

where the coordinate-wise conditional distribution (blue term) corresponds exactly to the minimizer of the training loss. Crucially, the first term of the right-hand side (green factor) does not depend on $\mathbf{x}_0$ and therefore cancels out during normalization, leaving the posterior unchanged. The same argument applies more generally as long as the forward process follows the form

$$p(\mathbf{x}_t = \mathbf{z} \mid \mathbf{x}_0, t) \propto \alpha_t(\mathbf{z}) \cdot \mathbf{1}\{\mathbf{x}_0^{\mathbf{um}(\mathbf{z})} = \mathbf{z}^{\mathbf{um}(\mathbf{z})}\}, \quad (2)$$

for an arbitrary nonnegative function $\alpha_t(\mathbf{z})$.

**Proposition 2** (Informal). *Under a forward process of the form of equation* (2)*, the training loss* (1)*'s unique minimizer does not change from the ground-truth unmasking posterior.*

Importantly, the teacher-forced chain has the form of equation (2): the selection of $\mathcal{S}$ depends only on the currently observed tokens in $\mathbf{x}_{t_j}$ (and time), as the unmasking policy $g_\phi(\mathbf{x}_{t_j}, t_j)$ does so, and never on the hidden entries of $\mathbf{x}_0$. Consequently, by Proposition 2, replacing the vanilla forward process with the teacher-forced chain changes only the *training distribution* while preserving the same unmasking-posterior minimizer of the MDM loss.

### 3.2. Empirical instantiation of PUMA

In this section, we describe how we instantiate PUMA in practice and provide pseudocode (Algorithm 1), focusing on the following implementation questions.

- How do we implement batch streaming for PUMA?

- Does PUMA introduce any computational overhead compared to standard MDM training?

**Streaming teacher-forced chains in a minibatch.** We implement PUMA as a streaming buffer of $B$ teacher-forced chains. Concretely, the minibatch state consists of tuples $\{(\mathbf{x}_0(j), \mathbf{z}(j), n(j))\}_{j=1}^B$, where $\mathbf{x}_0(j)$ is the underlying clean sample, $\mathbf{z}(j)$ is the current masked context (a state on its chain), and $n(j) \in \{0, \ldots, L/K\}$ is the current stage, where $L$ is the sequence length. At each training step, we compute the MDM loss on the current minibatch (one gradient step). We then advance *each* sequence to the next state of its chain (by revealing additional tokens according to PUMA). When a sequence reaches the end of the chain, we refresh it by drawing a new clean example $\mathbf{x}_0$, re-initializing a new chain. Thus, every intermediate state along each chain becomes a training sample. For prompt-based tasks, we keep the prompt unmasked at all stages and apply PUMA only to the non-prompt region.

**Policy from the current model.** As mentioned in Section 3.1, we instantiate the teacher-forced chain's $g_\phi$ on-the-fly using the current model itself. Specifically, given logits from $f_\theta(\cdot \mid \mathbf{z}(j))$, we compute a per-position confidence score for each masked index, e.g., the maximum predicted probability, rank masked positions by confidence, and unmask roughly $K$ tokens at each step (lines 8,9 in Algorithm 1). We say "roughly" because we found it empirically beneficial to add a little randomness to the exact number of unmasked tokens $K$, see Appendix B for details; and we also apply confidence-based fast-forwarding, see Section 3.2.1.

**No additional forward-pass overhead.** Importantly, PUMA does not introduce any extra compute: the logits required for the MDM loss on $\mathbf{z}(j)$ are exactly the quantities used to compute confidence scores.

#### 3.2.1. PRACTICAL INTERVENTIONS

Typically, MDM inference reveals only a small number of mask tokens per step. Directly mimicking this behavior at training with a small $K$ is sample-inefficient: a fixed $\mathbf{x}_0$ would generate a long teacher-forced chain, often providing a redundant training signal. We therefore introduce



*Figure 3.* **PUMA finds unmasking trajectories close to the final model's trajectories (right) early on in training.** We show PUMA training unmasking orders at different training steps for a given Sudoku puzzle. Green cells indicate prompt tokens.

two practical interventions that retain the guarantee of the teacher-forced chain while improving sample efficiency.

**Confidence-based fast-forwarding.** We *fast-forward* the chain if the next unmasking decisions are trivial for the current model. Concretely, at each step, we reveal roughly $K$ masked tokens selected by the policy but also reveal any masked position whose maximum predicted probability exceeds a confidence threshold, e.g., $0.9$. Intuitively, highly confident positions provide little training signal if kept masked, so revealing them early lets us follow the teacher-forced trajectory without stopping at every intermediate sequence.

$K$**-scheduling.** Early in training, the model-based policy is noisy, so using small $K$ can waste compute on long, low-quality chains. We therefore use $K$-*scheduling*: start with a large $K$ (yielding more frequent refreshes and greater diversity across $\mathbf{x}_0$) and gradually decrease $K$ as training progresses and the policy becomes more informative.

### 3.3. Sample complexity of PUMA

In this section, we provide a formal sample-complexity argument for a simple data distribution that shows that vanilla MDM training can suffer from exponential sample complexity, while PUMA under oracle trajectories avoids it.

**Distribution.** Motivated by the Latents-and-Observation framework of (Kim et al., 2025b), we consider a parametric subclass $p_\theta$ with learnable parameter $\theta$. Specifically, $p_\theta$ is a distribution over sequences $X = \pi(U_1, \ldots, U_d, Y) \in \mathcal{V}^L$, where $\mathcal{V}$ is a finite vocabulary, $\{U_i\}_{i=1}^d$ are latents, and $Y$ denotes observation. This distribution is defined through a fixed permutation $\pi \in S_L$ and parametrized by $\theta \in \Theta$ over a finite space $\Theta$. Writing $x = (u_1, \ldots, u_d, y)$, assume that the law of $p_\theta$ admits the following factorization: $p_\theta(\pi(x)) = p_U(u_{1:d})O_\theta(y \mid u_{1:d})$, where $p_U$ is a fixed distribution over $\mathcal{V}^d$ that does not depend on $\theta$, and $O_\theta(\cdot \mid u_{1:d})$ is a $\theta$-parameterized conditional distribution over $\mathcal{V}$. We further impose the following assumptions:

- *Posterior collapse:* There exists a fixed distribution $\pi_0$ on $\mathcal{V}$ such that for all $\theta \in \Theta$ and all strict subsets $S \subsetneq [d]$, the posterior of $p_\theta$ at index $\pi^{(L)}$, i.e. the $Y$-index, conditioned on $S$ is identical to $\pi_0$; in other words, all latents need to

be known in order to infer the observation.

- *Identifiability:* Let $P_\theta$ denote the joint law of $\pi(U_{1:d}, Y)$ under $p_\theta$. Assume there exists $\kappa > 0$ such that for all distinct $\theta, \theta' \in \Theta$, the *Chernoff information* between $P_\theta$ and $P_{\theta'}$ is at least $\kappa$; this condition ensures $\theta$ can be learned.

In Appendix A.3, we show that the above conditions are indeed easily fulfilled, e.g. by the additive group modulo $m$, alongside a certain choice of $p_\theta$.

Finally, we define the *oracle trajectory* as the unmasking sequence that reveals indices according to the permutation $\pi$, namely $\pi^{(1)}, \pi^{(2)}, \ldots$ (note that the specific order is immaterial, so long as the observation index $\pi^{(L)}$ is revealed last). We treat one complete execution of this trajectory as contributing $L$ training samples. Under the assumptions above, we can establish a sharp separation in sample complexity between oracle-guided and random unmasking. The formal statement and proof are deferred to Appendix A.3.

**Proposition 3** (Informal). *Choose an error threshold $\delta \in (0, 1)$. Under the above assumptions, random masking requires sample complexity exponential in $d$ to drive error below $\delta$, while the MAP estimator with PUMA on oracle trajectories only requires samples linear in $d$.*

Proposition 3 formalizes PUMA's training efficiency advantage. When equipped with the oracle trajectory $\pi$ – see Appendix A.3 for details on the connection between the oracle trajectories and PUMA's trajectories – PUMA requires only linear training samples, as it generates training samples that are aligned with the underlying structure of the distribution: latent variables are revealed earlier. In contrast, vanilla MDMs exhibit exponential sample complexity.

## 4. Experiments

In this section, we present experimental results for PUMA, focusing on two central claims:

- **Practicality**: PUMA speeds up MDM pretraining.

- **Compatibility**: PUMA offers complementary speedups on top of autoregressive initialization and block diffusion.

In Section 4.1, we validate our intuition of PUMA's design on Sudoku puzzles. In Section 4.2 and Section 4.5,

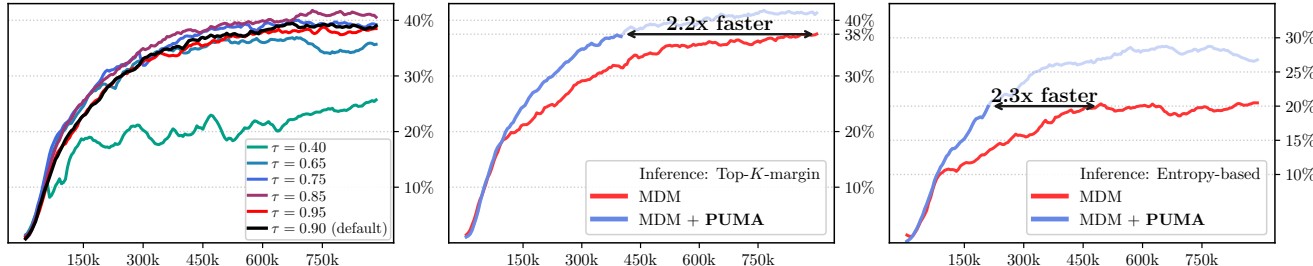

*Figure 4.* **Left**: PUMA's efficiency is largely insensitive to the confidence threshold, except when the threshold is very small. **Middle, Right**: A single PUMA-trained model (trained under one policy) remains robust to inference-time policy choices, consistently outperforming the baseline across different unmasking policies. (Top-K margin, Entropy)

we demonstrate PUMA's efficiency in 125M-scale MDM pretraining and in 7B-scale MDM post-training. Finally, in Section 4.4, we show that these speedups persist when PUMA is combined with other recipe-level remedies.

### 4.1. Sudoku puzzle as a testbed

In this section, we use Sudoku puzzles as a simple testbed to probe the practical viability of PUMA (recall Section 3.1).

**Setup.** We use the Sudoku dataset from Radcliffe (2020), implemented by Shah et al. (2024), with 1.8M training and 0.1M test puzzles. We train a 6.8M bidirectional transformer with Qwen2-style attention (Team et al., 2024), 8 layers, and hidden dimension 256. For PUMA, we use a fixed $K$-schedule with $K = 10$ and a confidence threshold of 0.9. Training is done on 2 NVIDIA H100 GPUs with a batch size of 32 per GPU, taking approximately an hour.

**Policy stabilization during training.** PUMA achieves 90.1% final accuracy (with the Top-2 unmasking rule and $|\mathcal{S}| = 2$), and speeds up training compared to the MDM baseline by $1.7\times$; Figure 5 in the appendix. Figure 3 visualizes PUMA's teacher-forced chain for a fixed instance at different checkpoints. At the early stage of training, the induced chain is uninformative. However, after a small number of iterations the pattern of the fully trained network's unmasking policy starts to emerge. In addition, the model becomes more confident in its predictions as training proceeds, and PUMA's confidence thresholding results in shorter chains. Figure 6 shows that the distance of training trajectories (formally defined in Appendix B) to the final model decreases quickly. This empirically validates that the *ranking* over positions (which drives the unmasking policy) can stabilize earlier than full posterior calibration (which drives the task accuracy), so using the current model for teacher-forced chains is a reasonable proxy already early in training.

### 4.2. PUMA accelerates pretraining

Moving beyond Sudoku puzzles, we test whether PUMA remains effective in a more realistic, larger-scale pretraining

regime. In choosing an appropriate scale and data domain, we aimed for a setup where (1) from-scratch pretrained models can achieve reasonable performance on real downstream benchmarks, e.g., GSM8K (Cobbe et al., 2021), while (2) the overall compute requirements remain feasible for running multiple controlled experiments.

**Dataset.** To satisfy both criteria, we adopt TinyGSM (Liu et al., 2023) as a pretraining corpus with 11.8M samples. TinyGSM converts the solutions of GSM8K-style math problems written in natural language into short and structured Python programs, making the task substantially more learnable at small model scales. Indeed, Liu et al. (2023) report that even a 125M-parameter autoregressive model pretrained from scratch on TinyGSM can reach reasonable GSM8K test set performance. Motivated by this, we use TinyGSM to study PUMA pretraining under a 125M MDM, which enables multiple runs and ablations under our compute budget.

**Setup.** We conduct an apples-to-apples comparison by measuring the *GSM8K test accuracy as a function of training iterations*. PUMA and the baseline differ *only* in batch control, but all other factors, including architecture, optimizer, and evaluation, are kept fixed. We use a 125M MDM with Qwen2-style attention (Team et al., 2024), with: hidden size 512, 14 layers, 8 heads, and maximum sequence length 512.

As stated in Section 3.2, we include $K$-scheduling and confidence-thresholding for PUMA. In particular, we initialize $K = 42$ and decrease it by 3 every 30k iterations until $K = 12$, and set the confidence threshold as 0.9. All runs train for 900k iterations, and evaluation is done on the standard GSM8K test set every 5k steps. We train on 8 NVIDIA H100 GPUs with batch size 32 per GPU, and training takes approximately three days. At inference, we unmask with the Top-2 scoring rule, using the maximum token-wise probability as the default confidence score. To improve robustness, we repeat each experiment three times with different random seeds. Furthermore, evaluation is done on EMA checkpoints as it often provides better accuracy for diffusion models (Karras et al., 2023).

**Results.** Figure 1 shows that PUMA accelerates (blue) MDM pretraining (red) by up to $2.3\times$ in terms of iterations-

to-accuracy. Noting that PUMA does not introduce additional forward-pass computation, this indicates that PUMA is a practically efficient training strategy.

We additionally measure wall-clock throughput: *3.78 it/s* for MDM versus *3.41 it/s* for PUMA. We attribute this small throughput gap to a loss function implementation detail. MDM uses the fused PyTorch primitive `F.cross_entropy`, whereas our PUMA implementation explicitly computes token log-probabilities (needed for the teacher-forced-chain update) and then forms the cross-entropy loss, which may incur extra kernel traffic.

**Robustness across inference policies.** PUMA's theoretical guarantee is established under the assumption that training and inference deploy the same unmasking policy. In practice, however, models are evaluated under a variety of policies. In light of this, we evaluate each training checkpoint under multiple unmasking policies. Specifically, we consider numerous sampling configurations, varying both $|\mathcal{S}|$ and inference-time unmasking policy, e.g., Top-K, Top-K margin, entropy-based, entropy-based thresholding, and non-zero temperature sampling. As shown in Figure 4 (Middle, Right) and Figure 9 in the appendix, PUMA consistently improves upon the baseline across all evaluation choices. This indicates that the resulting speedup transfers robustly to other unmasking strategies.

### 4.3. Ablations

We ablate two primary design choices of PUMA: *K scheduling* and *inference sampling strategy*. For $K$ scheduling, we compare our default run (with scheduling from $K = 42$ to $K = 12$) against a variant without $K$ schedule and a fixed value of $K = 12$. Furthermore, we provide an ablation across different $K$ schedules in Appendix B.3. Regarding the inference sampling strategy, we provide a sweep over the confidence threshold $\tau \in \{0.4, 0.65, 0.75, 0.85, 0.9 \text{ (default)}, 0.95\}$ in Figure 4, and a more extensive ablation over different inference strategies, including varying the number of decoded tokens per step and non-zero temperature sampling, in Appendix B.4.

**Results.** Figure 9 in the appendix shows that removing $K$ scheduling reduces the speed-up, indicating that the schedule is crucial for PUMA. In Figure 8 in Appendix B.3, we see that our default $K$ schedule outperforms other variants that decrease $K$ either more or less aggressively, suggesting that our $K$ schedule strikes a balance between the two. For confidence thresholding, Figure 4 suggests that PUMA's efficiency is broadly insensitive to the threshold, except when it is too low, such as 0.4 or 0.65. At such thresholds, PUMA unmasks numerous low-confidence positions which substantially alters the teacher-forced chain dynamics and degrades performance. Table 1 in Appendix B.4 shows that PUMA outperforms the MDM baseline uniformly across

all inference policies, and also tends to unmask around $10\%$ more tokens per step at inference under confidence-fast-forwarding inference policies, which we attribute to the fact that PUMA explicitly trains on inference-aligned trajectories, thus making it more familiar with the structured partially-masked states encountered at inference.

### 4.4. Combining PUMA with other remedies

In this section, we show that PUMA can be combined with common training recipes for MDMs at scale, demonstrating that PUMA's benefit is complementary to such remedies.

#### 4.4.1. AUTOREGRESSIVE MODEL INITIALIZATION

Initializing MDMs from pretrained autoregressive models is a well-known strategy for accelerating the training (He et al., 2023; Ye et al., 2025; Bie et al., 2025; Gong et al., 2025; DeepMind, 2025; Fan et al., 2026). To this end, we first pretrain an autoregressive model with the same transformer architecture as in Section 4.2 for 200k steps, which takes approximately 9.5 hours on 8 NVIDIA H100 GPUs. We then initialize the MDM from this checkpoint, change to bidirectional attention, and train it with and without PUMA. We keep other configurations the same as in Section 4.2. Figure 1 reports the resulting learning curves (purple vs. green). While both methods benefit from ARM initialization, PUMA exhibits a substantially larger gain, achieving a speedup of approximately $4.0\times$ over the baseline.

#### 4.4.2. BLOCK-SIZE WARMUP

Block diffusion (Arriola et al., 2025a;b) combines block-wise autoregressive generation with MDM-style inference within each block, with reported improved training efficiency over vanilla MDM training. Motivated by this, recent large-scale pretraining efforts (Bie et al., 2025; Fan et al., 2026) adopt a *block-size warmup*: MDM is initialized with a pretrained autoregressive model (block size equals the full sequence length) and gradually transitions toward an MDM. Empirically, this warmup improves training efficiency.

We train block diffusion models using a block size 256, and aggregate the training loss across blocks for each sample. For PUMA, we generate a teacher-forced chain as in Section 3.2, but restricted to each block. As shown in Figure 9, PUMA also accelerates training in this setting, which shows that its benefits are complementary to block diffusion.

### 4.5. Applying PUMA to 7B-scale MDM

In this section, we demonstrate that PUMA stays effective in post-training of pretrained MDMs, which has been shown to be crucial for downstream performance (Ye et al., 2025; Gong et al., 2025; Bie et al., 2025).

**Setup.** We collect 700k training pairs from opc-sft-stage-2-educational (Huang et al., 2024), KodCode-V1-SFT-4o (Xu et al., 2025), and OpenCodeInstruct (Ahmad et al., 2025), while padding all sequences to a maximum length of 640. We then fine-tune Dream-Coder-Base-7B (Xie et al., 2025) with a frozen backbone and LoRA (Hu et al., 2022) rank 128, resulting in approximately 300M trainable parameters. We compare vanilla training against PUMA, using a fixed $K$-schedule with $K = 15$ and a confidence threshold of 0.9. All runs are conducted on 8 H100 GPUs with a global batch size of 256. Following the exact evaluation protocol of Dream-Coder's public codebase, we report Pass@1 accuracy on HumanEval (Chen et al., 2021) and MBPP (Austin et al., 2021), evaluated every 2,000 training steps.

**Results.** As shown in Figure 7, PUMA yields clear gains over vanilla MDM fine-tuning on both benchmarks. On HumanEval, PUMA improves performance by about 10 percentage points, whereas vanilla MDM tuning fluctuates and achieves at most a 3% gain. On MBPP, vanilla tuning again exhibits larger fluctuations, while PUMA delivers a consistent improvement of approximately 5 percentage points. This shows that PUMA is beneficial even in large-scale LoRA fine-tuning.

**Remark.** We hypothesize that the relatively modest gain of vanilla fine-tuning stems from the fact that LoRA restricts optimization to a smaller trainable subspace, making high-quality training signals especially important. In this regard, PUMA is particularly well-suited to this setup, as it focuses training on inference-aligned predictions.

## 5. Discussion

**Discussion on evidence lower bound.** The evidence lower bound (ELBO) is one clear way to derive the loss objective of MDMs (Sahoo et al., 2024; Shi et al., 2024), and having a reliable ELBO can be important in several practical setups, e.g., reinforcement learning. This raises a question of whether an ELBO guarantee still holds for models trained with PUMA. We empirically compare the resulting ELBOs in our TinyGSM and Dream-Coder fine-tuning experiments and defer a more detailed discussion to Appendix B.6.

**Empirical comparison to PAPL (Peng et al., 2025b).** We empirically compare PUMA with PAPL (Peng et al., 2025b) on Sudoku and TinyGSM. We sweep PAPL's reweighting hyperparameter $\alpha \in \{1, 3, 5\}$ for Sudoku, and use the default $\alpha = 1$ for TinyGSM. PAPL uses the same random masking procedure as vanilla MDM training, but reweights each cross-entropy term in the training loss proportional to the model's confidence, scaled by $\alpha$.

As shown in Appendix B.5, PAPL does not help in training on either of our datasets, and often even hurts performance; this trend is consistent across the MDM baseline and PUMA,

as well as different choices of $\alpha$. This shows that when performance is sensitive to the inference policy, as in Sudoku and TinyGSM, directly changing the forward process can be substantially more effective than loss reweighting alone. That said, PAPL might still be preferable in domains that are less sensitive to generation order, or in setups where using only a single forward pass per training sample is important, since PUMA requires multiple forward passes.

**Connection to self-forcing (Huang et al., 2025).** Although different in technical detail, our method is conceptually related to self-forcing (Huang et al., 2025). In self-forcing, a diffusion model performs inference-style rollout during training to reduce the train–inference gap. At a high level, this is close to our teacher-forced chain, which also trains the model under the inference-time policy.

## 6. Conclusion

**Limitation.** Our empirical evaluation of PUMA, while already covering meaningful settings, is still limited to 125M-scale pretraining and 7B-scale post-training. Another limitation of PUMA is that it cannot be implemented efficiently in a single-epoch setup, since the teacher-forced chain naturally entails multiple forward passes for a single sequence. That said, our experimental results challenge the view that a single forward pass per sample is necessarily preferable even in such a regime; we consistently observe that during the first epoch, PUMA outperforms the baseline when compared at the same number of gradient steps.

**Broader Perspective.** Accelerating diffusion model training has been extensively studied in continuous domains. While these approaches explore different objectives and training principles, they cannot operate beyond the *standard forward-process* framework: Gaussian noise is added independently across dimensions. As a result, most existing degrees of freedom take the form of modifying a scalar noise schedule, reweighting the loss (Nichol & Dhariwal, 2021; Salimans & Ho, 2022), or introducing an auxiliary regularizer (Yu et al., 2024).

Masked diffusions, in contrast, admit a far wider design space. They admit substantial flexibility in the *decoding order–which positions are revealed when*. Prior work (Zheng et al., 2024; Kim et al., 2025b) has shown that this flexibility can be exploited at inference time. This flexibility is a defining feature of masked diffusions and lies squarely outside the scope of continuous diffusion formulations.

In this work, we show that this structural freedom can also be exploited at training time. PUMA leverages the discrete nature of masked diffusion by modifying the forward masking process to align the distributions over masked sequences induced at train- and inference-time, opening the door to a new design axis for masked diffusion models.

## Acknowledgments

SC was supported in part by NSF CAREER award CCF-2441635. DAM acknowledges support from the Kempner Institute, NSF award no. 2229881, and the Aramont Fellowship Fund. JG is supported by a Kempner Graduate Fellowship. SK acknowledges support from the Kempner Institute.

## Impact Statement

This paper presents work whose goal is to advance the field of Machine Learning. There are many potential societal consequences of our work, none of which we feel must be specifically highlighted here.

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

# A. Theory

## A.1. Proof of Proposition 1

In this section, we provide the complete proof of Proposition 1.

**Clarification of the policy notation.** In the main text, we write $\mathcal{S} = g_\phi(\mathbf{z}, t)$ to denote the *output* of the unmasking policy, which in practice can be stochastic. In this appendix, we adopt the equivalent distributional view under the simplifying assumption $|\mathcal{S}| = 1$: namely, we write $g_\phi(i \mid \mathbf{z}, t)$ for the conditional probability of selecting the next index $i$ given the current state $(\mathbf{z}, t)$. Sampling $I \sim g_\phi(\cdot \mid \mathbf{z}, t)$ and setting $\mathcal{S} = \{I\}$ recovers the main-text notation.

**Notation.** For $\mathbf{z} \in \mathcal{Z} := (\mathcal{V} \cup \{\mathbf{m}\})^L$, define the unmasked and masked index sets $\mathrm{um}(\mathbf{z}) := \{i \in [L] : \mathbf{z}_i \neq \mathbf{m}\}$ and $\mathrm{msk}(\mathbf{z}) := [L] \setminus \mathrm{um}(\mathbf{z})$. For $i \in [L]$ and $v \in \mathcal{V}$, write $z^{(i \leftarrow v)} \in \mathcal{Z}$ for the sequence obtained by replacing the $i$-th entry of $z$ with $v$ (leaving all other coordinates unchanged).

Let $\mathbf{x}_0 \sim p_{\mathrm{data}}$ be a clean sequence in $\mathcal{V}^L$. For any partially-masked $\mathbf{z} \in \mathcal{Z}$ and any $i \in \mathrm{msk}(\mathbf{z})$, the (time-agnostic) ground-truth unmasking posterior is $p(\mathbf{x}_0^i = v \mid \mathbf{z}) := \mathbb{P}(\mathbf{x}_0^i = v \mid \mathbf{x}_0^{\mathrm{um}(\mathbf{z})} = \mathbf{z}^{\mathrm{um}(\mathbf{z})})$.

We now restate the formal version of Proposition 1.

**Proposition 1** (Formal). *Fix an integer $l \geq 1$ and a time grid $t_j := 1 - j/l$ for $j = 0, 1, \ldots, l$. Consider the following two Markov chains on $\mathcal{Z}$, both initialized at the fully masked state $(m, \ldots, m)$:*

1. *Given $\mathbf{z}_{t_j}$, sample an index $I_j \sim g_\phi(\cdot \mid \mathbf{z}_{t_j}, t_j)$ (so $I_j \in \mathrm{msk}(\mathbf{z}_{t_j})$), then sample a token $V_j \sim p(\mathbf{x}_0^{I_j} = \cdot \mid \mathbf{z}_{t_j})$, and set $\mathbf{z}_{t_{j+1}} := \mathbf{z}_{t_j}^{(I_j \leftarrow V_j)}$. Let $q_{t_j}$ denote the law of $\mathbf{z}_{t_j}$ under this procedure.*

2. *First sample $\mathbf{x}_0 \sim p_{\mathrm{data}}$ and set $\tilde{\mathbf{z}}_{t_0} = (m, \ldots, m)$. Given $\tilde{\mathbf{z}}_{t_j}$, sample $I_j \sim g_\phi(\cdot \mid \tilde{\mathbf{z}}_{t_j}, t_j)$ and set $\tilde{\mathbf{z}}_{t_{j+1}} := \tilde{\mathbf{z}}_{t_j}^{(I_j \leftarrow \mathbf{x}_0^{I_j})}$. Let $\tilde{q}_{t_j}$ denote the marginal law of $\tilde{\mathbf{z}}_{t_j}$ after sampling $\mathbf{x}_0 \sim p_{\mathrm{data}}$.*

*Then, for every $j \in \{0, 1, \ldots, l\}$, we have $q_{t_j} = \tilde{q}_{t_j}$.*

*Proof.* We show that the two chains have the same one-step transition kernel from every state $\mathbf{z} \in \mathcal{Z}$.

Fix a step $j$ and a state $\mathbf{z} \in \mathcal{Z}$. Under the teacher-forced chain, the event $\{\tilde{\mathbf{z}}_{t_j} = \mathbf{z}\}$ is possible only if the sampled clean sequence $\mathbf{x}_0$ matches $\mathbf{z}$ on the revealed coordinates, i.e., $\mathbf{x}_0^{\mathrm{um}(\mathbf{z})} = \mathbf{z}_{\mathrm{um}(\mathbf{z})}$. Moreover, conditioned on $\mathbf{x}_0^{\mathrm{um}(\mathbf{z})} = \mathbf{z}_{\mathrm{um}(\mathbf{z})}$, the probability of realizing $\tilde{\mathbf{z}}_{t_j} = \mathbf{z}$ depends only on the policy randomness along the trajectory (since $g_\phi$ reacts only to the currently revealed tokens and $t$), and is therefore identical for all $\mathbf{x}_0$ that agree with $\mathbf{z}$ on $\mathrm{um}(\mathbf{z})$. Consequently, for some scalar $\alpha_j(\mathbf{z}) \geq 0$ and all $\mathbf{x} \in V^L$,

$$\mathbb{P}(\tilde{\mathbf{z}}_{t_j} = \mathbf{z} \mid \mathbf{x}_0 = \mathbf{x}) = \alpha_j(\mathbf{z})\,\mathbf{1}\{\mathbf{x}_{\mathrm{um}(\mathbf{z})} = \mathbf{z}_{\mathrm{um}(\mathbf{z})}\}.$$

By Bayes' rule, this implies

$$\mathbb{P}(\mathbf{x}_0 = \mathbf{x} \mid \tilde{\mathbf{z}}_{t_j} = \mathbf{z}) \propto p_{\mathrm{data}}(\mathbf{x})\,\mathbf{1}\{\mathbf{x}_{\mathrm{um}(\mathbf{z})} = \mathbf{z}_{\mathrm{um}(\mathbf{z})}\}.$$

In particular, for any masked coordinate $i \in \mathrm{msk}(\mathbf{z})$ and any $v \in V$, $\mathbb{P}(\mathbf{x}_0^i = v \mid \tilde{\mathbf{z}}_{t_j} = \mathbf{z}) = p(x_0^i = v \mid \mathbf{z})$ by the definition of the ground-truth unmasking posterior.

Now fix $i \in \mathrm{msk}(\mathbf{z})$ and $v \in V$. Using the above conditional identity and the fact that $I_j$ is drawn from $g_\phi(\cdot \mid \mathbf{z}, t_j)$ given the current state, the teacher-forced transition probability is

$$\mathbb{P}\Big(\tilde{\mathbf{z}}_{t_{j+1}} = \mathbf{z}^{(i \leftarrow v)} \mid \tilde{\mathbf{z}}_{t_j} = \mathbf{z}\Big) = g_\phi(i \mid \mathbf{z}, t_j)\,\mathbb{P}(\mathbf{x}_0^i = v \mid \tilde{\mathbf{z}}_{t_j} = \mathbf{z}) = g_\phi(i \mid \mathbf{z}, t_j)\,p(x_0^i = v \mid \mathbf{z}).$$

On the other hand, the idealized inference chain satisfies

$$\mathbb{P}\Big(\mathbf{z}_{t_{j+1}} = \mathbf{z}^{(i \leftarrow v)} \mid \mathbf{z}_{t_j} = \mathbf{z}\Big) = g_\phi(i \mid \mathbf{z}, t_j)\,p(x_0^i = v \mid \mathbf{z}),$$

since it samples the same index distribution and then draws the revealed token from the same posterior. Therefore, the two chains have identical transition kernels and the same initial condition $\mathbf{z}_{t_0} = \tilde{\mathbf{z}}_{t_0} = (m, \ldots, m)$. It follows by induction on $j$ that $q_{t_j} = \tilde{q}_{t_j}$ for all $j \in \{0, 1, \ldots, l\}$, and $\mathbb{P}(\mathbf{x}_0^i = v \mid \tilde{\mathbf{z}}_{t_j} = \mathbf{z}) = p(\mathbf{x}_0^i = v \mid \mathbf{z})$ by the definition of the ground-truth unmasking posterior. $\qquad\square$

**Remark (continuous-time limit).** The discrete-time proof above is stated on a grid and under the one-index-per-step idealization. In the continuous-time limit $l \to \infty$, the corresponding dynamics converge to a continuous-time Markov chain on $\mathcal{Z}$. The same argument applies at the level of rate matrices, which can be understood as the *infinitesimal transition kernel*: conditioned on the current partially revealed state, the next revealed token in the teacher-forced construction has the same conditional distribution as sampling from the ground-truth posterior, hence the two generators coincide. Therefore, the marginal agreement extends to all $t \in [0, 1]$ in the continuous-time formulation.

### A.2. Proof of Proposition 2

This section contains the formal version of Proposition 2 and its proof.

We start with a general lemma about the minimizers of weighted cross-entropy loss functions.

**Lemma 4** (Weighted cross-entropy is minimized by the true conditional). *Let $(X, Y)$ be any jointly distributed random variables where $Y$ takes values in a finite set $\mathcal{V}$. Let $w(X) \geq 0$ be any measurable function.*

*Consider the functional*

$$\mathcal{L}(q) := \mathbb{E}\big[w(X)\,(-\log q(Y \mid X))\big], \tag{3}$$

*where $q(\cdot \mid x)$ ranges over all conditional distributions on $\mathcal{V}$.*

*Then any minimizer $q^\star$ satisfies*

$$q^\star(\cdot \mid x) = \mathbb{P}(Y = \cdot \mid X = x) \tag{4}$$

*for $\mathbb{P}$-almost every $x$ such that $w(x) > 0$. If $w(x) > 0$ holds $\mathbb{P}$-almost surely, the minimizer is unique up to $\mathbb{P}$-a.s. equality.*

*Proof.* Conditioning on $X = x$,

$$\mathbb{E}[-\log q(Y \mid X) \mid X = x] = H\big(\mathbb{P}(Y = \cdot \mid X = x)\big) + \mathrm{KL}\big(\mathbb{P}(Y = \cdot \mid X = x) \,\|\, q(\cdot \mid x)\big). \tag{5}$$

Multiply by $w(x) \geq 0$ and take expectations over $X$. The entropy term is independent of $q$, while the KL term is nonnegative and equals 0 iff $q(\cdot \mid x) = \mathbb{P}(Y = \cdot \mid X = x)$. This yields the claim and uniqueness on $\{w > 0\}$. $\qquad\square$

**Proposition 2** (Formal). *Let the masking process take the form*

$$p(\mathbf{x}_t = \mathbf{z} \mid \mathbf{x}_0, t) \propto \alpha_t(\mathbf{z}) \cdot \mathbf{1}\{\mathbf{x}_0^{\mathbf{um}(\mathbf{z})} = \mathbf{z}^{\mathbf{um}(\mathbf{z})}\},$$

*where $\alpha_t : (\mathcal{V} \cup \{\mathbf{m}\})^L \to [0, 1]$ is a weighting function. Denote the density over $(x_0, t, x_t)$ induced by this forward process by $p_f$, i.e.*

$$p_f(x_0, t, x_t) = p(x_t | x_0, t) p_{\mathrm{data}}(x_0) \mathbf{1}_{[0,1]}(t),$$

*where $\mathbf{1}_{[0,1]}$ is the density of the uniform distribution over $[0, 1]$. Then any minimizer $f_\theta^*$ of the training loss*

$$\mathcal{L}(\theta) = \mathbb{E}_{(\mathbf{x}_0, t, \mathbf{x}_t) \sim p_f}\left[\frac{1}{t} \sum_{i\,:\,\mathbf{x}_t^i = \mathbf{m}} -\log f_\theta^i(\mathbf{x}_0^i \mid \mathbf{x}_t)\right]$$

*is the ground-truth posterior of $p_{\mathrm{data}}$, and this posterior is equal to the distribution conditioned on the unmasked indices of $x_0$ and $x_t$ coinciding, i.e.*

$$f_\theta^*(x_0 \mid x_t) = \mathbb{P}(x_0 \mid x_t) \propto \mathbf{1}\{x_0^{\mathbf{um}(x_t)} = x_t^{\mathbf{um}(x_t)}\}\, p_{\mathrm{data}}(x_0).$$

*Proof.* We start by showing the equality

$$\mathbb{P}(x_0 \mid x_t) \propto \mathbf{1}\{x_0^{\mathbf{um}(x_t)} = x_t^{\mathbf{um}(x_t)}\}\, p_{\mathrm{data}}(x_0).$$

Fix $t \in (0, 1)$ and a masked context $x_t \in (\mathcal{V} \cup \{\mathbf{m}\})^L$. Write $\mathbf{um}(x_t) \subseteq [L]$ for the unmasked coordinates and $\mathbf{msk}(x_t) := \{i : x_t^i = \mathbf{m}\}$. By assumption,

$$p(x_t \mid x_0, t) \propto \alpha_t(x_t)\, \mathbf{1}\{x_0^{\mathbf{um}(x_t)} = x_t^{\mathbf{um}(x_t)}\}. \tag{6}$$

Then Bayes' rule gives, for any $x_0$,

$$\mathbb{P}(x_0 \mid x_t, t) \propto p(x_t \mid x_0, t)\, p_{\text{data}}(x_0) \propto \alpha_t(x_t)\, \mathbf{1}\{x_0^{\mathbf{um}(x_t)} = x_t^{\mathbf{um}(x_t)}\}\, p_{\text{data}}(x_0),$$

The factor $\alpha_t(x_t)$ does not depend on $x_0$ and cancels after normalization. Furthermore, the dependency on $t$ also disappears, yielding the *data conditional* form

$$\mathbb{P}(x_0 \mid x_t) = \mathbb{P}(x_0 \mid x_t, t) \propto \mathbf{1}\{x_0^{\mathbf{um}(x_t)} = x_t^{\mathbf{um}(x_t)}\}\, p_{\text{data}}(x_0). \tag{7}$$

In particular, the right-hand side depends on $x_t$ only through its revealed coordinates, and thus does not depend on $\alpha_t$.

Now let $(\mathbf{x}_0, t, \mathbf{x}_t) \sim p_f$. Fix a coordinate $i \in [L]$ and define the random variables

$$X := \mathbf{x}_t, \qquad Y := \mathbf{x}_0^i \in \mathcal{V}.$$

The contribution of coordinate $i$ to the loss is

$$\mathcal{L}_i(\theta) := \mathbb{E}_{X,Y,t}\Big[\frac{1}{t}\, \mathbf{1}\{X^i = \mathbf{m}\}\, \big(-\log f_\theta^i(Y \mid X)\big)\Big].$$

By equation (7), we have $\mathbb{P}(Y = \cdot \mid X, t) = \mathbb{P}(Y = \cdot \mid X)$. Therefore, we can rewrite

$$\mathcal{L}_i(\theta) = \mathbb{E}_{X,Y}\left[\mathbb{E}_t\left[\frac{1}{t}\, \mathbf{1}\{X^i = \mathbf{m}\}|X\right]\big(-\log f_\theta^i(Y \mid X)\big)\right]$$

$$= \mathbb{E}_{X,Y}\Big[w_i(X)\big(-\log f_\theta^i(Y \mid X)\big)\Big],$$

where $w_i(X) := \mathbb{E}_t\big[\frac{1}{t}\, \mathbf{1}\{X^i = \mathbf{m}\}|X\big] \geq 0$ (note that we are assuming $t > 0$). Thus, by Lemma 4, any minimizer $f_\theta^*$ equals the true posterior $\mathbb{P}(x_0 \mid x_t)$, which finishes the proof. $\qquad\square$

### A.3. Proof of Proposition 3

In this section, we provide the proof of Proposition 3. We start off with a Chernoff bound lemma.

**Lemma 5.** *Let $P, Q$ be distributions on a finite space and let $Z_1, \ldots, Z_T \sim P$ i.i.d. Then*

$$\mathbb{P}\Big(\prod_{t=1}^T \frac{Q(Z_t)}{P(Z_t)} \geq 1\Big) \leq e^{-TC(P,Q)},$$

*where $C(P, Q)$ is the Chernoff information between $P$ and $Q$.*

*Proof.* Recall that

$$C(P, Q) := -\log\Big(\inf_{s \in (0,1)} \sum_z P(z)^{1-s} Q(z)^s\Big).$$

For any $s \in (0, 1)$, Markov's inequality and independence of the $Z_t$ gives

$$\mathbb{P}\Big(\prod_{t=1}^T \frac{Q(Z_t)}{P(Z_t)} \geq 1\Big) = \mathbb{P}\Big(\Big(\prod_{t=1}^T \frac{Q(Z_t)}{P(Z_t)}\Big)^s \geq 1\Big) \leq \mathbb{E}\Big[\prod_{t=1}^T \Big(\frac{Q(Z_t)}{P(Z_t)}\Big)^s\Big] = \Big(\sum_z P(z)^{1-s} Q(z)^s\Big)^T.$$

Minimizing over $s \in (0, 1)$ yields the claim. $\qquad\square$

For clarity, we restate our two assumptions, *posterior collapse* and *identifiability*.

**Assumption 6** (Posterior collapse). There exists a fixed distribution $\pi_0$ on $\mathcal{V}$ such that for all $\theta \in \Theta$ and all strict subsets $S \subsetneq [d]$, the posterior of $p_\theta$ at index $\pi^{(L)}$, i.e. the $Y$-index, conditioned on $S$ is identical to $\pi_0$; in other words, all latents need to be known in order to infer the observation.

**Assumption 7** (Identifiability). Let $P_\theta$ denote the joint law of $\pi(U_{1:d}, Y)$ under $p_\theta$. Assume there exists $\kappa > 0$ such that for all distinct $\theta, \theta' \in \Theta$, the *Chernoff information* between $P_\theta$ and $P_{\theta'}$ is at least $\kappa$; this condition ensures $\theta$ can be learned.

Now we can prove Proposition 3.

**Proposition 3** (Formal)**.** *Let $\delta \in (0, 1)$, and denote by $q$ the masking probability under random masking. Under the above assumptions, random masking requires at least $n = \Omega(q^{-1}(1 - q)^{-d} \log(1/\delta))$ training samples to drive error below $\delta$, whereas $n_{\mathrm{traj}} = O((d + 1) \log(|\Theta|/\delta)/\kappa)$ samples suffice for the MAP estimator when training with PUMA on oracle trajectories. For fixed $q$, the dependence $q^{-1}(1 - q)^{-d}$ is exponential in $d$.*

*Proof.* First, we prove the exponential bound for random masking. Denote by $F$ the event that all latents $U_{1:d}$ are unmasked, and $Y$ is masked, in the random masking setting. Then $\mathbb{P}(F) = q(1 - q)^d$, hence $\mathbb{P}(\neg F) = 1 - q(1 - q)^d$. Hence, the probability of $n$ i.i.d. masking samples all being equal to $\neg F$ is $(1 - q(1 - q)^d)^n$. Note that conditioned on $\neg F$, the distribution $p_\theta$ is identical across all $\theta \in \Theta$, since $p_U$ (i.e. the distribution of the latents) is independent of $\theta$ by assumption, and the conditional distribution of $Y$ is independent of $\theta$ by Assumption 6. Hence, if all samples are masked as $\neg F$, no estimator can recover $\theta$ better than random chance over $\Theta$. Thus,

$$\mathbb{P}(\widehat{\theta} \neq \theta) \geq \left(1 - \frac{1}{|\Theta|}\right)\left(1 - q(1 - q)^d\right)^n,$$

and $(1 - a)^n \geq e^{-an}$ yields

$$\mathbb{P}(\widehat{\theta} \neq \theta) \geq \left(1 - \frac{1}{|\Theta|}\right)\exp\left(-nq(1 - q)^d\right).$$

In particular, to make $\mathbb{P}(\widehat{\theta} \neq \theta) \leq \delta$ it is necessary that

$$n \geq \frac{1}{q(1 - q)^d} \log\left(\frac{1 - 1/|\Theta|}{\delta}\right).$$

Next, we prove the linear bound for masking with oracle trajectories. Let $\theta$ be the data generating parameter, and $Z_1, ..., Z_T \sim P_\theta$, where $T$ is the number of samples drawn from the distribution, i.e. the number of training samples (counted as the number of steps in trajectories) equals $n_{\mathrm{traj}} = LT$. We now consider the $Z_i$ to be the *informative* training steps in the oracle trajectories, i.e. where all $U_i$ are revealed and $Y$ is masked. This has the same distribution as $P_\theta$, so we can apply the Chernoff lower bound from Assumption 7. Let $\theta' \neq \theta$ be a wrong hypothesis. Each of the $T$ oracle trajectories contains one step where all the latents are unmasked and $Y$ is masked; The event that $\theta'$ has likelihood at least $\theta$ implies

$$\prod_{t=1}^{T} \frac{P_{\theta'}(Z_t)}{P_\theta(Z_t)} \geq 1.$$

By Lemma 5 and Assumption 7,

$$\mathbb{P}(\text{MAP selects } \theta' \text{ instead of } \theta) \leq e^{-TC(P_\theta, P_{\theta'})} \leq e^{-T\kappa}.$$

A union bound over the $|\Theta| - 1$ wrong parameters yields

$$\mathbb{P}(\theta' \neq \theta) \leq (|\Theta| - 1)\, e^{-T\kappa}.$$

In particular, to ensure that $\mathbb{P}(\theta' \neq \theta) \leq \delta$ it suffices to take

$$T \geq \frac{1}{\kappa}\left(\log(|\Theta| - 1) + \log(1/\delta)\right),$$

which gives

$$n_{\mathrm{traj}} = (d + 1)T = O\left((d + 1)\frac{\log(|\Theta|/\delta)}{\kappa}\right). \qquad \square$$

*Remark* 8*.* For simplicity, we defined $p_\theta$ over a fixed permutation $\pi$. However, the above results can also be extended to a dataset where permutations can vary, e.g. by being drawn from a predefined small set of permutations.

Next, we give a simple example distribution that shows that the setting and assumptions above can indeed be instantiated in practice, and that ground-truth confidence-unmasking yields oracle trajectories as above in practice.

**Example 9.** Let $m \geq 4$ be even, $\mathcal{V} = \mathbb{Z}_m$, and set $\Delta := m/2$. Take $\Theta = \{0, \Delta\}$ and $d \geq 1$. Draw latents $U_1, \ldots, U_d \overset{i.i.d.}{\sim}$ $\mathrm{Unif}(\{0, \Delta\})$ and independent noise $E$ with $\mathbb{P}(E = 0) = 1 - \eta$ and $\mathbb{P}(E = a) = \eta/(m-1)$ for $a \neq 0$, where $\eta \in (0, 1/2)$. Define

$$Y := \theta + \sum_{j=1}^{d} U_j + E \pmod{m}.$$

*Collapse.* For any strict $S \subsetneq [d]$, the sum of the missing latents $\sum_{j \notin S} U_j$ is supported on $\{0, \Delta\}$ and its law is invariant under shifts by $\Delta$; hence $Y \mid U_S$ has the same law for $\theta = 0$ and $\theta = \Delta$. Thus Assumption 6 holds (with $\pi_0$ equal to this common conditional law).

*Separability.* At the informative step ($U_{1:d}$ revealed, $Y$ masked in the input), the observed pair $(U_{1:d}, Y)$ has two hypotheses that differ by shifting the noise by $\Delta$. Evaluating the Chernoff coefficient at $s = 1/2$ gives

$$C(P_\theta^{\mathrm{inf}}, P_{\theta'}^{\mathrm{inf}}) \geq -\log B =: \kappa > 0, \quad B = 2\sqrt{(1 - \eta)\frac{\eta}{m-1}} + (m-2)\frac{\eta}{m-1},$$

thus Assumption 7 holds as well.

*Ground-truth confidence order.* Each latent has distribution $\mathrm{Unif}(\{0, \Delta\})$, so the ground truth confidence for each latent, defined as the highest probability of any token in $\mathcal{V}$ for that latent, is $1/2$; since latents are independent, this does not change when some latents have already been revealed. If at least one latent is missing, then $Y$ equals a (known shift of) $E$ convolved with $\mathrm{Unif}(\{0, \Delta\})$, so $\mathrm{conf}(Y) = \frac{1}{2}(1 - \eta) + \frac{1}{2}\frac{\eta}{m-1} < 1/2$. Once all latents are revealed, the confidence of $Y$ is $1 - \eta > 1/2$ instead; hence any rule that unmasks positions with highest ground-truth confidence reveals latents before $Y$.

## B. Training Details and Experiments

### B.1. Omitted details of PUMA.

The overall PUMA pipeline follows Algorithm 1, with one detail on how the number of newly unmasked tokens $|\mathcal{S}_j|$ is determined at each step.

**Summarization.** The basic idea behind the design is twofold: First, we aim to unmask a roughly uniform number of tokens $K$ at each stage of the diffusion process, while introducing controlled stochasticity in the exact number of tokens unmasked. This stochasticity is important in practice: deterministically unmasking the same number of tokens at every stage can lead to an imbalance in the distribution of masking patterns encountered during training.

Second, in addition to stage-based unmasking, we optionally apply confidence-based thresholding, which further unmasks tokens whose confidence scores exceed a fixed threshold (see Section 3.2.1).

**Stage partitioning.** Fix a $K$ such that $L/K$ is an integer. We partition the interval $[0, 1]$ into $L/K$ uniform subintervals $I_\ell := \left[\frac{\ell}{L/K}, \frac{\ell+1}{L/K}\right]$ for $\ell = 0, \ldots, L/K - 1$. For an intermediate masked sequence $\mathbf{x}_t$, we define its *stage* $n \in \{0, \ldots, L/K - 1\}$ based on the fraction of unmasked tokens. Concretely, $\mathbf{x}_t$ is said to be at stage $n$ if the number of unmasked (i.e., clean) tokens lies in the range $\left[L_{\mathrm{eff}} \cdot \frac{n}{L/K}, L_{\mathrm{eff}} \cdot \frac{n+1}{L/K}\right]$, where $L_{\mathrm{eff}}$ denotes the effective sequence length, i.e., the number of tokens in $\mathbf{x}_0$ excluding prompt tokens.

**Teacher-forced chain movement.** Assume we are given a tuple $(\mathbf{x}_t, n, \mathbf{x}_0)$ consisting of the current masked sequence, its stage index, and the corresponding clean sequence. To advance the teacher-forced chain from stage $n$ to (approximately) stage $n + 1$, we proceed as follows:

1. Sample target unmasking ratio: Sample $r \sim \mathrm{Unif}(I_{n+1})$, which specifies the desired fraction of unmasked tokens for the next stage.

2. Determine the number of tokens to unmask: Let $U_t$ denote the number of currently unmasked tokens in $\mathbf{x}_t$. The target number of unmasked tokens is $L_{\mathrm{eff}} \cdot r$, and hence the number of newly unmasked tokens is $\Delta U = L_{\mathrm{eff}} \cdot r - U_t$.

3. Generate the next state: We unmask $\Delta U$ tokens according to the PUMA unmasking policy, obtaining the next intermediate sequence, which we nominally associate with stage $n + 1$.

**Interaction with confidence thresholding.** When confidence thresholding is enabled, we additionally unmask all tokens whose confidence scores exceed a fixed threshold $\tau$. Since this step may unmask more tokens than prescribed by $\Delta U$, the total number of unmasked tokens can exceed $L_{\text{eff}} \cdot r$. In this case, after all unmasking operations are completed, we recompute the stage index based on the final number of unmasked tokens, ensuring that the stage assignment remains consistent with the definition above.

**Training configuration.** Table 2 shows our training and evaluation hyperparameters for both our Sudoku and TinyGSM experiments. All hyperparameters are identical across PUMA and the MDM baseline (except for PUMA-specific parameters, such as the $K$-schedule, which do not apply to the baseline).

### B.2. Additional experimental results

**Discussion on the block diffusion experiment.** Block diffusion additionally *predefines* the inference-time unmasking order (block-by-block). This rigidity can partially counteract PUMA's speed-up, which in part comes from aligning the training-time masking process with the inference-time unmasking policy. Consequently, while we do not sweep over block sizes, we expect PUMA's incremental gain to decrease as the block size becomes smaller (i.e., as the inference order becomes more constrained).

Importantly, our main claim (Section 4.4) is that PUMA is complementary to block-size warmup: the warmup schedule necessarily passes through a regime with relatively large blocks, and we expect this is precisely the regime where PUMA's policy-alignment effect applies most strongly.

**Sudoku training speedup.** Figure 5 shows the training speedup achieved by PUMA over the MDM baseline in our training run from Section 4.1.

**Sudoku training trajectories.** Figure 6 shows the distance from the training trajectories found by PUMA over the course of training to the trajectories found by the final model. Here, the distance at training time $t$ is defined as

$$d_{\text{abs}}(t, T) := \mathbb{E}_{x_0} \left[ \frac{1}{81} \sum_{(i,j) \in [9] \times [9]} \left| u_{ij}^t(x_0) - u_{ij}^T(x_0) \right| \right], \tag{8}$$

where $T$ is the time at the end of training, and $u_{ij}^t$ is the integer *step* in the trajectory at which cell $(i,j)$ is unmasked; e.g., if a cell is unmasked at the $k$th step in the trajectory, $u_{ij}^t = k$. In Figure 6, we average over 100 (fixed) samples. The figure shows that the distance decreases rapidly early during training, which validates using the online trajectories from the current model for teacher-forced chains. However, interestingly, the distance saturates at around 0.5, which means a small gap in trajectory distance persists.

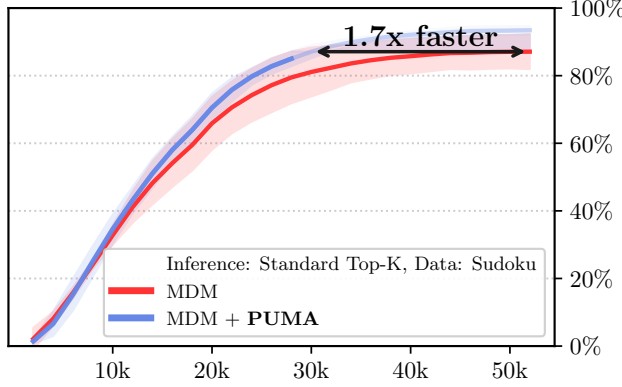

*Figure 5.* **PUMA speeds up training on Sudoku by** $1.7\times$**.**

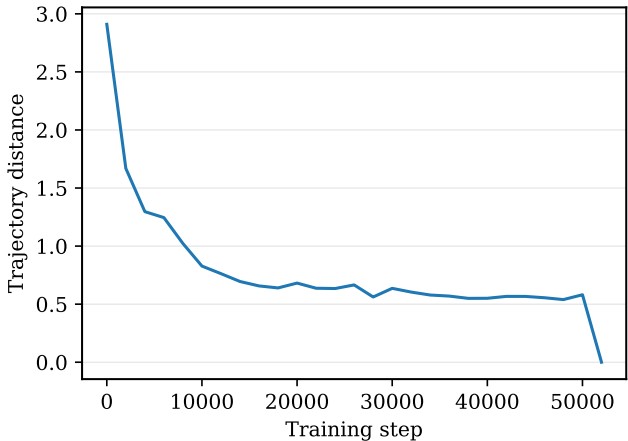

*Figure 6.* **Training trajectory distance to the fully trained model diminishes quickly.** We show the distance (8) between the training trajectory found by PUMA over the course of training, averaged over a fixed set of 100 samples on Sudoku.

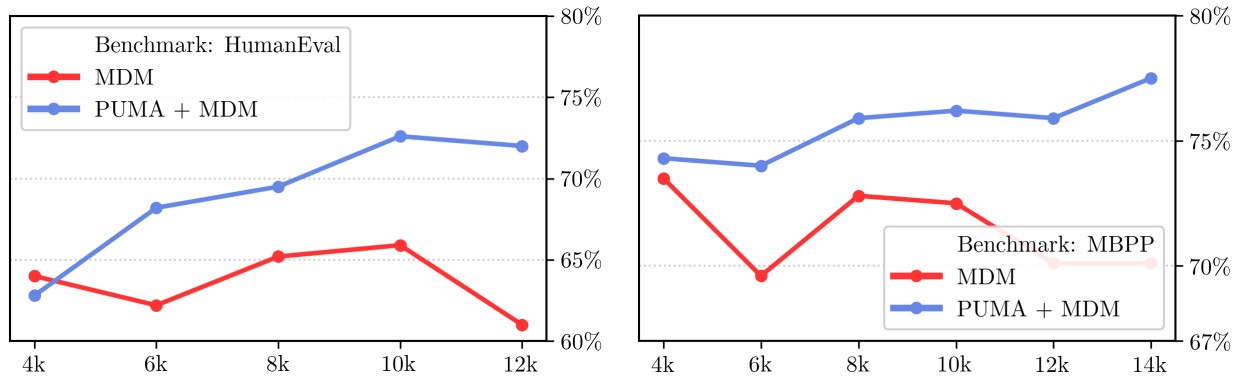

*Figure 7.* **PUMA outperforms the MDM baseline on Dream-Coder fine-tuning across benchmarks.**

### B.3. $K$ schedule ablation

In this section, we provide an ablation over the $K$ schedule on TinyGSM. In Figure 8, we compare our $K$ schedule to other variants, including some that decrease $K$ more or less aggressively than our schedule. We can see that while most schedules are competitive, our standard schedule has a slight edge. Furthermore, schedules that decrease $K$ too aggressively hurt performance noticeably.

### B.4. Ablation over inference-time unmasking policies

In Table 1, we compare PUMA against the MDM baseline for various inference policies. This includes confidence-fast-forwarding policies, where, akin to training, we unmask all tokens above a certain confidence threshold in each step (falling back to the two highest-confidence tokens if none are above the threshold); varying values of $K$ in the Top-$K$ inference policy; and temperature sampling, where in each step, tokens are sampled according to the specified temperature in all positions, and the top two tokens according to model confidence are unmasked.

The results show that PUMA uniformly outperforms the MDM baseline across all inference strategies; we note that some strategies lead to bad performance of both methods, such as very large values of $K$, which is to be expected. Furthermore, Table 1 shows that for the confidence-fast-forwarding strategies, where the number $K$ of tokens unmasked in each step is not fixed, PUMA tends to unmask approximately $10\%$ more tokens per step than the MDM baseline. We attribute this to the fact that PUMA explicitly trains on inference-aligned trajectories, which should make PUMA more familiar with the structured partially-masked states encountered at inference, leading to higher confidences overall.

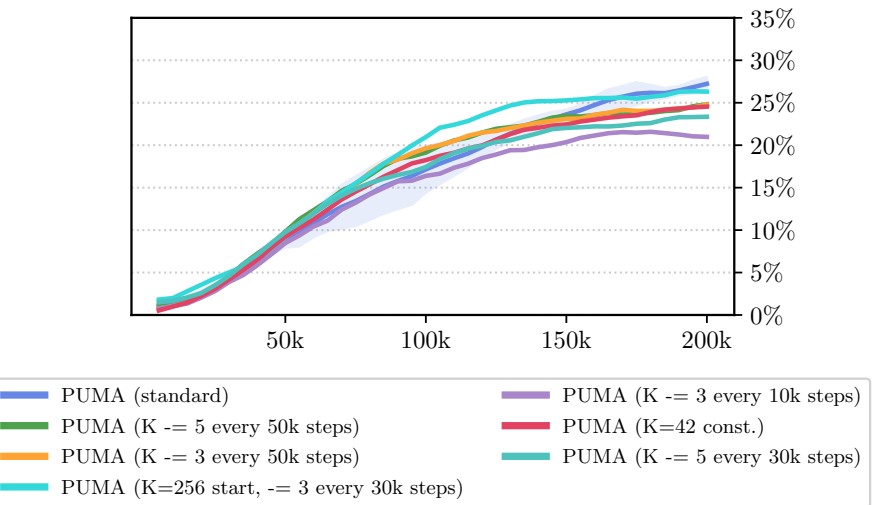

*Figure 8.* **Our standard $K$-schedule outperforms other $K$-schedule variants.** The standard $K$-schedule starts at 42 and decreases by 3 every 30k steps, down to 12. Each schedule starts at 42, except for the one that explicitly notes $K = 256$ at start.

## B.5. Empirical comparison to PAPL (Peng et al., 2025b)

In this section, we define the PAPL training objective and empirically compare it to PUMA and the MDM baseline on both Sudoku and TinyGSM. Recall the MDM training loss:

$$\mathcal{L}(\theta) = \mathbb{E}_{(\mathbf{x}_0,t,\mathbf{x}_t)} \left[ \frac{1}{t} \sum_{i:\, \mathbf{x}_t^i = \mathbf{m}} -\log f_\theta^i(\mathbf{x}_0^i \mid \mathbf{x}_t) \right].$$

PAPL reweights the same masked cross-entropy terms according to the model's confidence in the correct token at each masked position. In our notation, define

$$w_t^i = \frac{f_\theta^i(\mathbf{x}_0^i|\mathbf{x}_t)}{\sum_{j:\, \mathbf{x}_t^j = \mathbf{m}} f_\theta^j(\mathbf{x}_0^j|\mathbf{x}_t)},$$

so that positions where the denoiser already assigns higher probability to the clean token receive larger weight. The PAPL objective can then be written as

$$\mathcal{L}_{\text{PAPL}}(\theta) = \mathbb{E}_{(\mathbf{x}_0,t,\mathbf{x}_t)} \left[ \frac{1}{t} \sum_{i:\, \mathbf{x}_t^i = \mathbf{m}} \left( 1 + \alpha w_t^i \right) \left( -\log f_\theta^i(\mathbf{x}_0^i \mid \mathbf{x}_t) \right) \right],$$

where $\alpha > 0$ controls the strength of the planner-aware reweighting. The authors recommend $\alpha = 1$ as a default, but sometimes observe improved performance for values up to $\alpha = 5$; increasing $\alpha$ beyond 5 generally hurts performance.

In Figure 10, we compare PAPL against the MDM baseline and PUMA on both Sudoku and TinyGSM, including a sweep across $\alpha = 1,\ 3,\ 5$ on Sudoku. Across both datasets and training paradigms, we observe that PAPL does not improve training efficiency, and oftentimes even seems to hurt.

## B.6. Discussion on evidence lower bound

We evaluated the MDM NELBO on our TinyGSM validation dataset (236k samples) with PUMA and the MDM baseline. Figure 11, left, shows the result. The vanilla MDM training attains a lower NELBO than PUMA, and this is explainable as the NELBO equals the training objective of the baseline. However, even though the NELBO does not exactly align with the PUMA training objective, PUMA is reasonably good at minimizing the NELBO as well, and attains a final NELBO of around 0.06, while the MDM baseline attains 0.05.

Moreover, we conduct the same comparison on Dream-Coder-7B, with a 35k validation set and three random time steps per sequence. The result is shown in Figure 11, right. Here, we observe a different trend: PUMA yields a lower NELBO than

*Table 1.* **PUMA consistently outperforms the MDM baseline across all inference policies.** We ablate (a) the confidence threshold $\tau$ in confidence-fast-forwarded inference (akin to training), (b) the number of unmasked tokens $K$, and (c) temperature sampling, where we first sample tokens according to the specified temperature and then unmask the top two tokens based on their confidence scores. Tokens/step are reported where $K$ is not fixed.

| Ablation | Setting | Accuracy | | Tokens / step | |
|---|---|---|---|---|---|
| | | PUMA | MDM Baseline | PUMA | MDM Baseline |
| **Confidence fast-forward** | $\tau = 0.80$ | 36.3 | 31.0 | 5.3 | 5.0 |
| | $\tau = 0.90$ | 37.5 | 33.1 | 3.8 | 3.4 |
| | $\tau = 0.95$ | 36.9 | 32.8 | 3.3 | 3.1 |
| **Top-K** | $K = 2$ | 39.1 | 33.6 | – | – |
| | $K = 5$ | 37.5 | 33.5 | – | – |
| | $K = 10$ | 28.0 | 22.4 | – | – |
| | $K = 20$ | 14.1 | 8.6 | – | – |
| **Temperature top-2** | $T = 0.2$ | 37.5 | 33.0 | – | – |
| | $T = 0.5$ | 37.0 | 32.8 | – | – |
| | $T = 0.7$ | 37.8 | 31.8 | – | – |
| | $T = 0.8$ | 37.6 | 33.7 | – | – |
| | $T = 1.0$ | 39.5 | 33.4 | – | – |

vanilla fine-tuning. We hypothesize this is driven by two related factors. First, while PUMA unmasks tokens in *easy* orders, our confidence-fast-forwarded training ensures that easy positions are unmasked early, so the model can spend more time on hard tokens. Second, random unmasking might spend compute on patterns that the model has already learned successfully. Both of these factors can lead to more meaningful gradient updates for PUMA and could explain the lower NELBO.

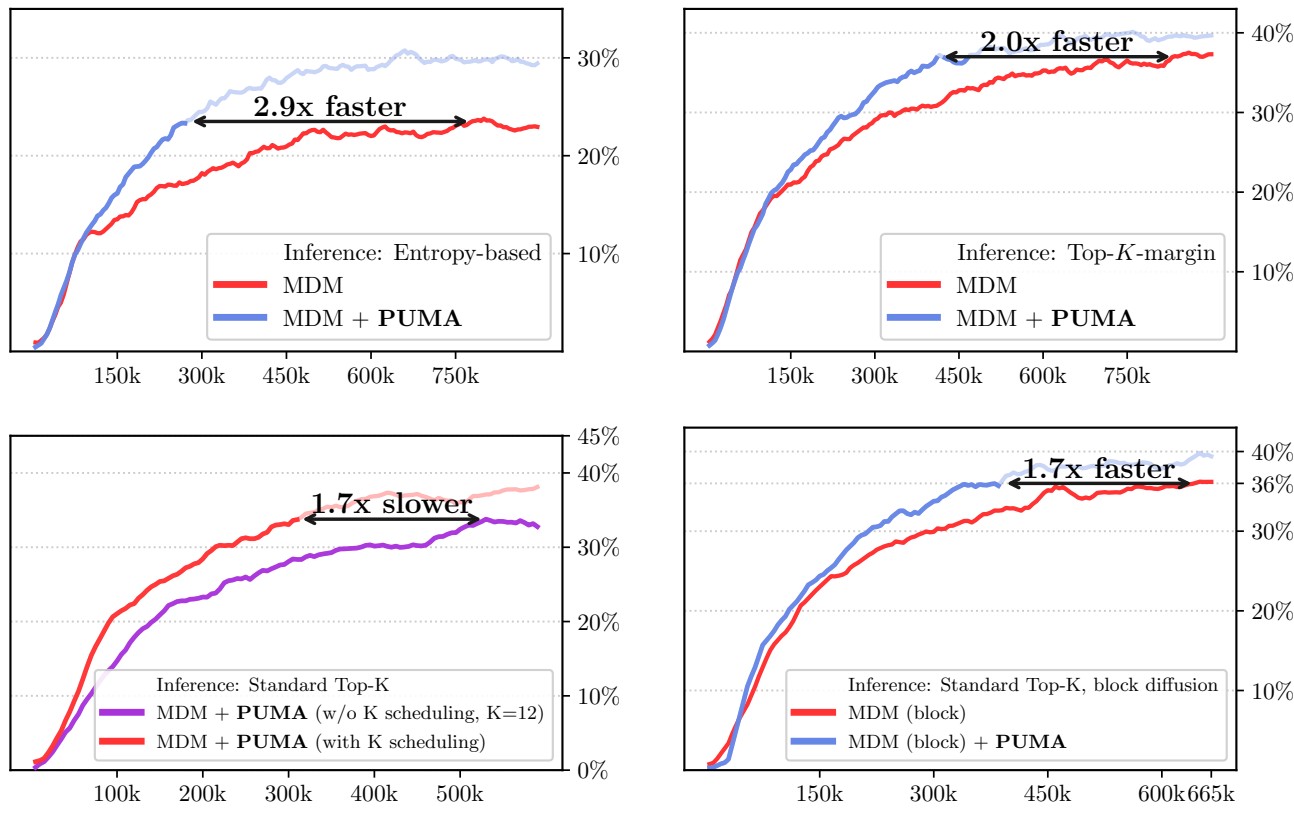

*Figure 9.* **Lower Left**: $K$ scheduling is an important factor for PUMA's efficiency; removing scheduling (fixed $K = 12$) degrades iteration-accuracy efficiency. (purple curve). **Upper**: A single PUMA-trained model (trained under one policy) remains robust to inference-time policy choices, consistently outperforming the baseline across different unmasking policies. Figure 4 shows the iteration-accuracy curves with $|S| = 2$. Here, we present the curves with $|S| = 3$ under the Top-K margin (Upper Right) and entropy-based (Upper Left) policy. **Lower Right**: PUMA also yields speedup for block diffusion training with block size 256.

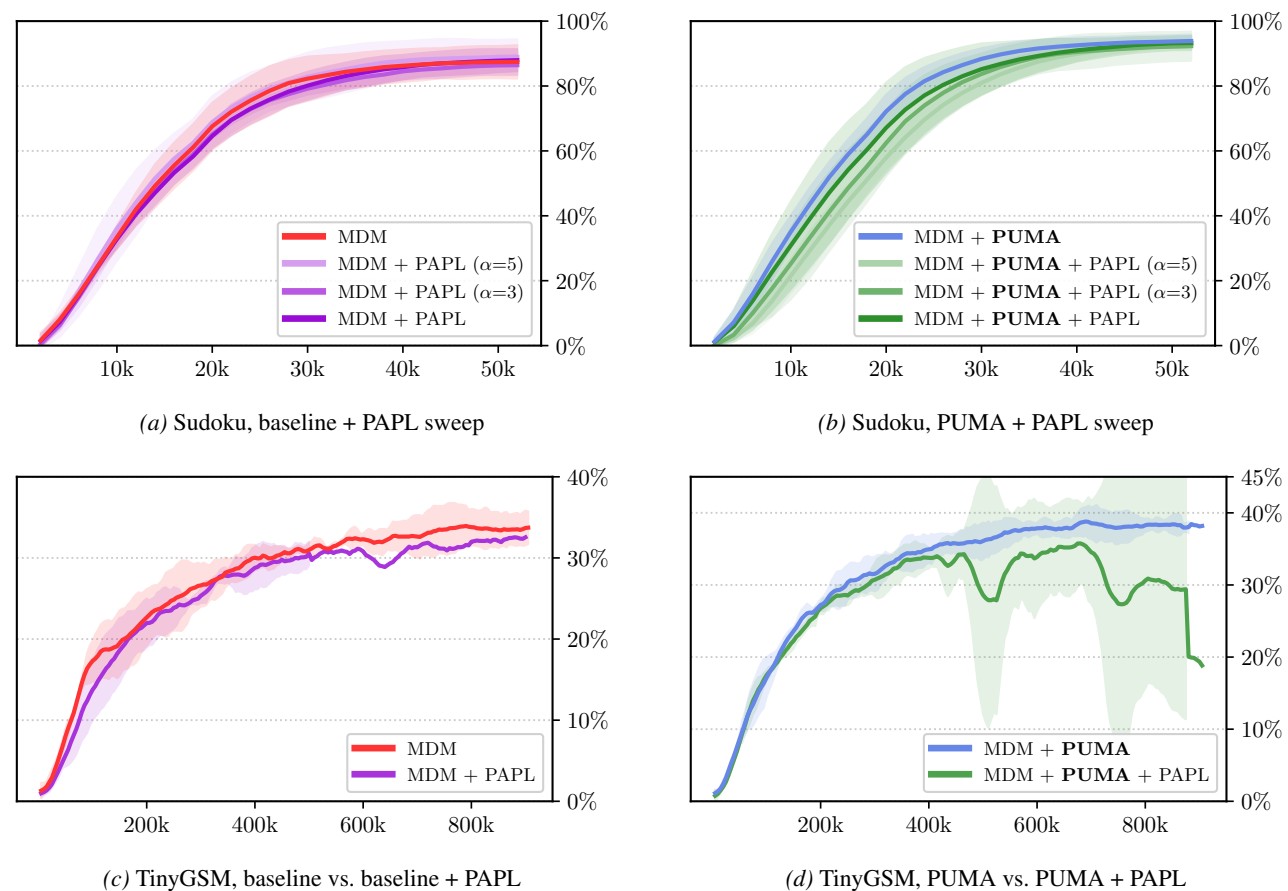

*(a)* Sudoku, baseline + PAPL sweep

*(b)* Sudoku, PUMA + PAPL sweep

*(c)* TinyGSM, baseline vs. baseline + PAPL

*(d)* TinyGSM, PUMA vs. PUMA + PAPL

*Figure 10.* **PAPL does not improve MDM training on Sudoku or TinyGSM.** This is consistent for both the MDM baseline and PUMA models. We plot 95% CIs across three random seeds. We note that the poor performance of PAPL on TinyGSM with PUMA was caused by one of the three seeds; the other two performed just slightly worse than PUMA.

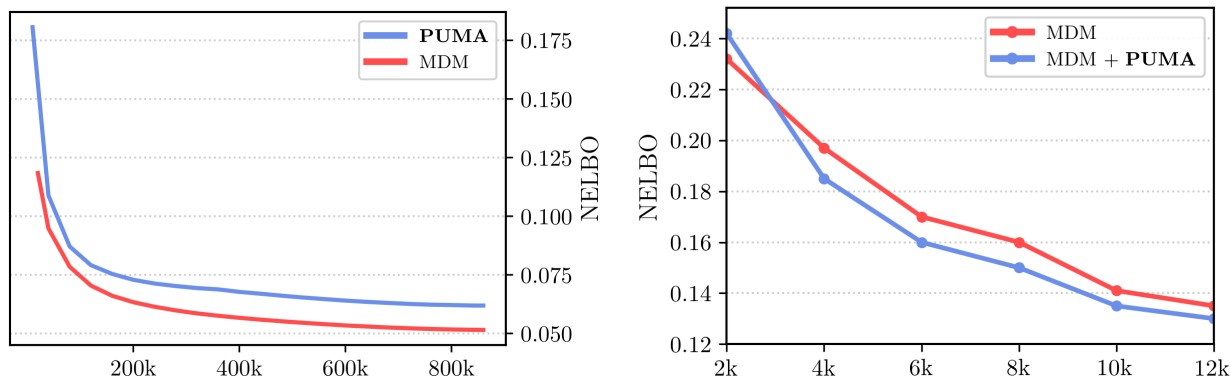

*Figure 11.* **Left:** NELBO comparison on TinyGSM. **Right:** NELBO comparison on Dream-Coder fine-tuning.

*Table 2.* Training and evaluation hyperparameters on Sudoku and TinyGSM.

|  | **Sudoku** | **TinyGSM** |
| --- | --- | --- |
| Tokenizer | Custom | Qwen/Qwen2-0.5B |
| Vocab size | 10 | 151645 |
| Model size | 6.8M | 125M |
| Hidden dim | 256 | 512 |
| MLP dim | 768 | 1536 |
| Number of layers | 8 | 14 |
| Number of heads | 8 | 8 |
| Max length | 81 | 512 |
| RMSNorm epsilon | 1e−6 | 1e−6 |
| Dropout | 0 | 0 |
| Confidence threshold | 0.9 | 0.9 |
| K schedule | 10 (fixed) | 42–12 (in decrements of 3 every 30k steps) |
| Learning rate | 3e−4 | 3e−4 |
| Warmup steps | 1000 | 1000 |
| EMA value | 0.9999 | 0.9999 |
| Weight decay | 0.01 | 0.01 |
| Epochs | 2 | 20 |
| Max grad norm | 1 | 1 |
| Batch size (per GPU) | 32 | 32 |
| Number of GPUs | 2 | 8 |
| Sampling temperature | 0.0 | 0.0 |

