# OpenReview forum: "Stop Training for the Worst: Progressive Unmasking Accelerates Masked Diffusion Training"
_ICML.cc/2026/Conference — ICML 2026 regular_

### Official Review · Reviewer_S6bL · 2026-03-01

**Soundness:** 3
**Presentation:** 2
**Significance:** 4
**Originality:** 3
**Overall Recommendation:** 4
**Confidence:** 4

**Summary:**

This paper introduces Progressive UnMAsking (PUMA), a straightforward adjustment to the forward masking process that harmonizes the masking patterns used during training and inference. By doing so, PUMA focuses optimization on inference-aligned masks, thereby accelerating training. At the 125M parameter scale, PUMA significantly expedites pretraining and delivers complementary benefits when combined with common techniques such as autoregressive initialization.

**Compliance With Llm Reviewing Policy:**

Affirmed.

**Key Questions For Authors:**

Please see the weakness.

**Limitations:**

yes.

**Strengths And Weaknesses:**

Strengths:

1. This paper optimizes the training process of DLLM by addressing the training-inference mismatch problem in diffusion language models, significantly accelerating pretraining. This makes perfect sense and is insightful.

2. The paper meticulously designs the training process, ensuring no extra forward computation is wasted during training, maintaining training efficiency and making better use of each training sample.

3. Experimental results strongly demonstrate the effectiveness of the proposed method.

4. This method is also applicable to both AR-initialized DLLM and block diffusion.

Weaknesses:

1. The proposed method is only validated on a 125M model and limited pretraining data. However, considering the high cost of pretraining, this limitation is acceptable.

2. Why does PUMA only improve training speed but not significantly improve the final model's performance? Intuitively, PUMA should also improve model accuracy.

3. Will the PUMA-trained model encounter problems when using parallel decoding with a confidence threshold? Because most samples encountered during training may have masks appearing more on the left side of the sequence and less on the right side, this could impair the model's ability to perform parallel decoding.

---

> ### Author Rebuttal · Authors · 2026-03-31
>
> We are grateful to the reviewer for their positive feedback to our paper and appreciate how they note it being **”insightful”** and **”strongly demonstrat[ing] performance”**. We address their questions below.
>
> > **The proposed method is only validated on a 125M model**
>
> Although the reviewer noted that the limited scaling in our paper is understandable given the compute constraints, we would like to highlight that we demonstrate that **PUMA is applicable to 7B-scale fine-tuning** in a novel suite of experiments. We refer you to the “Experiment Setup” paragraph in our response to reviewer **knjU** for the details. In summary, on HumanEval, PUMA yields about a 10% gain within 12k iterations, whereas vanilla MDM tuning fluctuates and achieves at most a 3% gain. On MBPP, vanilla tuning is even less stable, while PUMA delivers a consistent ~5% improvement. This shows that **for large-scale LoRA finetuning, the gains of PUMA over the baseline are even more pronounced** than in the pretraining setting. We attribute this to the fact that LoRA provides a smaller parameter space to optimize, which makes it even more important to provide a rich training signal, which PUMA achieves by limiting training to inference-aligned predictions. The results are shown here: https://ibb.co/5xjs7sRq.
>
> > **Why does PUMA only improve training speed but not significantly improve the final model's performance?**
>
> In principle, as noted in Proposition 2, **PUMA does not change the ground-truth minimizer of the training loss**, which would imply performance on par with the baseline at optimality. In practice, however, an MDM cannot perfectly realize this minimizer at every unmasking step, so the empirical unmasking posterior still differs from the ground-truth one. As a result, the inference trajectory can significantly affect performance. From this perspective, PUMA improves accuracy by aligning training more closely with the posteriors encountered at inference time. That being said, because both training recipes share the same minimizer, **the empirical gap between PUMA and the vanilla MDM can depend on the domain**. For example, in more structured settings such as Sudoku, vanilla MDM can achieve similar accuracy, whereas in HumanEval, where the search space is much larger, vanilla fine-tuning does not reach the same performance, even with substantially longer SFT.
>
> > **Because most samples encountered during training may have masks appearing more on the left side of the sequence and less on the right side, this could impair the model's ability to perform parallel decoding?**
>
> We thank the reviewer for this question and would appreciate some clarification. We are unsure why, during training, masks would appear “more on the left side” of the sequence and less on the right side. If the reviewer is referring to masks resembling causal autoregressive patterns, those would typically place masks more on the right side, not the left. We would also appreciate clarification on the concern regarding parallel decoding. In our evaluation, we already decode multiple tokens at a time, as described in the paper, and our results indicate that this works well. It would therefore be helpful if the reviewer could elaborate on the specific issue they have in mind.

---

> > ### Author Rebuttal · Reviewer_S6bL · 2026-04-03
> >
> > Given a confidence threshold of 0.9, would the PUMA-trained model result in a lower Token Per Forward than the MDM-trained model?

---

> > > ### Author Response · Authors · 2026-04-07
> > >
> > > > **Given a confidence threshold of 0.9, would the PUMA-trained model result in a lower Token Per Forward than the MDM-trained model?**
> > >
> > > We have added evaluations of both PUMA and the MDM baseline with confidence thresholds of 0.8, 0.9, and 0.95. In particular, in each inference step, we unmask all tokens above the threshold. If there are none, we instead unmask the top-2 confidence tokens.
> > >
> > >
> > > | **Method** | **PUMA acc.** | **PUMA tok/step** | **MDM acc.** | **MDM tok/step** |
> > > |---|---|---|---|---|
> > > | **τ = 0.80** | 36.3 | 5.3 | 31.0 | 5.0 |
> > > | **τ = 0.90** | 37.2 | 3.8 | 33.3 | 3.4 |
> > > | **τ = 0.95** | 36.9 | 3.3 | 32.8 | 3.1 |
> > >
> > > As can be seen from the results, **PUMA significantly outperforms the MDM baseline in terms of accuracy across all thresholds, while unmasking roughly 10% more tokens in each step**. We attribute this to the fact that **PUMA explicitly trains on inference-aligned trajectories, which should make PUMA more familiar with the structured partially-masked states encountered at inference**, leading to higher confidences overall. For more detailed evaluation results, including temperature-based decoding, we refer you to our second response to reviewer **n54q**.
> > >
> > > We thank the reviewer for taking the time to review our paper. **If the reviewer feels that their questions have been properly addressed, we kindly ask them to consider raising their score.**

---

### Official Review · Reviewer_n54q · 2026-03-08

**Soundness:** 2
**Presentation:** 3
**Significance:** 2
**Originality:** 3
**Overall Recommendation:** 5
**Confidence:** 5

**Summary:**

This paper introduce Progressive Unmasking (PUMA) a method to accelerate training of masked diffusion models. The goal of PUMA is to align the training-time masking patterns with the inference-time unmasking policy (e.g. entropy, top 2 margin). Concretely, PUMA is implemented using a buffer that holds the training examples, and after performing a training step on example $x$ in the buffer, the tokens where the denoising neural network is most confident are un-masked, and replaced by the clean token values. The buffer is then updated to use this lower-noise level sequence in further steps. Crucially, this assumes the model is trained repeatedly on the same sequences, *which is the case int the current, small scale, training settings studied in academia*. In the best training-from-scratch experiments, PUMA allows reaching a fixed accuracy after $2.5x$ fewer steps than standard MDM training. The training throughput between standard MDM and PUMA is roughly the same, since the denoiser predictions are used to update the training example for the next training step.

**Compliance With Llm Reviewing Policy:**

Affirmed.

**Final Justification:**

The authors have addressed my concerns. Updating their manuscript is crucial for acceptance, but I believe they will do it.

**Key Questions For Authors:**

My questions follow directly from the weaknesses. For completeness, the questions are stated concisely here, and sorted in order of importance.

- What happens in the multi-epoch setting?
- Intuition for why the K-curriculum is needed?
- How sensitive is PUMA to the specific hyper-parameters of the K-curriculum?
- How can we compute an ELBO with PUMA? How does it compare to the standard MDM NELBO?
- What happens with more realistic AR initialization?
- How is the few / many sampling-step performance?
- Are you using stratified time selection?
- What is the performance of an AR baseline trained on tinyGSM ?
- How is the zero-shot likelihood?
- (Experimental details). What context size are you using? What are you doing with too-long training examples? Are you wrapping / padding the training examples?
- What happens with non-zero temperature sampling?

**Limitations:**

As stated in the weaknesses section, the limitations are not sufficiently discussed. Most importantly, PUMA assumes a multi-epoch training, which is not the current practice of LLM pre-training. This is the most important limitation in my opinion but is not discussed.

**Strengths And Weaknesses:**

## Strengths
- The paper is well-written, and easy to follow. The colored text helps readability, and is used sparingly to highlight relevant elements.
- The motivation is well-supported, and the authors show that PUMA keeps the minimizer of the loss unchanged compared to standard MDM training. The argument compares the Bayes-optimal denoiser under the standard MDM training and PUMA, and shows these are equal.
- PUMA introduces negligible training overhead, *assuming the training data is used repeatedly*.
- PUMA is compatible with block diffusion and AR initialization
- The authors evaluate on GSM8K, using the TinyGSM training set, instead of relying on the Generative Perplexity, which is commonly used in other discrete diffusion papers, but can be gamed by generating low diversity samples. This provides a better signal for ablations, and I believe most future papers on discrete diffusion will adopt this dataset as their main evaluation benchmark.


## Weaknesses
Despite the strengths of the paper, *there are several weaknesses that are crucial to address*. The paper studies an important question in masked diffusion training, however **the discussion of the limitations is not sufficient**. In the following, I first discuss the most pressing issues with the current paper. I believe these must be addressed before the paper can be accepted.


### Crucial weaknesses
- **PUMA is only applicable in the multi-epoch setting**: since PUMA unmasks tokens based on the confidence of the denoiser during the previous use of a training example $x$, **PUMA is efficient only in the multi-epoch setting**. In the case of TinyGSM, this is not a problem, however, in industry, when training LLMs, it is common to **not** repeatedly see the same training examples. In this case, PUMA is *NOT* applicable as-is, and would require two forward passes. **This weakness is not discussed at all**. Furthermore, there are no experiments to study what happens in such setting. Namely, a possible fix would be to use PUMA for a certain fraction of the steps, eg 20% of the steps, and standard MDM training otherwise. In such case, does PUMA still bring training-time advantages? Such multi-epoch setting can be simulated by *not* using a buffer, and acting as if every training batch is fresh. **This is the most important weakness of PUMA that needs to be studied in priority**. Indeed, this would provide some sign on whether PUMA can be applied in large-scale training, or whether it is only useful in small-scale settings.
- **PUMA assumes a discrete time discretization and does not experiment with few-steps sampling**. The authors do not discuss how the accuracy varies when using more or fewer sampling steps than seen during training. In particular, this is important to compare against an MDM trained with the standard loss in this setting as well.
- **The AR initialization is not realistic**: in the AR initialization experiment, the authors first train an AR model for 30k steps, and then initialize an MDM using the AR weights. This setting is not realistic, because prior work (e.g. DiffuLlama, Gong et al) initializes the MDM from a **strong** AR model. From the manuscript, it is not clear whether twhat he gains would be if the MDM was initialized from an AR checkpoint trained eg for 500k steps. Recall that the authors train for up to 900k steps in total, hence an initialization after 30k steps of AR training is very little.
- **Intuition for the K-curriculum**: empirically, the authors find that using a fixed, pre-determined number of noise levels (denoted as K) is detrimental (1.7x **slower** training) compared to standard MDM training (Figure 6, lower right). To address this issue, the authors use a warmup, where the number of noise levels $K$ is progressively increased. First, there are no ablations on how fast $K$ should be increased. **Importantly, it is not clear how sensitive is the training gains to the choice of K**. This is important to understand, because **if the procedure is sensitive to particular value of $K$ or warmup schedule, it is unlikely to be adopted in practice**. Finally, do you have any intuition why the curriculum is needed? This is not answered by your theoretical results I believe.
- **Connection between PUMA and the MDM NELBO**: prior work directly optimize the NELBO, which induces a weighted cross-entropy loss. The authors do not discuss how to approximate the likelihood of a given sequence $x$ after training with PUMA. For example, how does the likelihood compares after training with PUMA, if evaluated with the standard MDM NELBO? This is important for tasks such as RL fine-tuning. Furthermore, can we derive an alternative NELBO assuming the PUMA forward process?

### Other weaknesses
- **Missing experimental details**: the authors do not describe the data preparation sufficiently. What context size is being used? Are you packing training examples (i.e. concatenating all examples, and separating them with EOS tokens) ? This is important because for efficiency, this is a common trick used during pre-training. If PUMA requires padded sequences to work, this significantly reduces the practicality of PUMA. Additionally, are you discarding training examples that are too long and do not fit in the context length?
- **Are the authors using stratified time selection?** Many recent work on discrete diffusion use *stratified* time sampling during training. Concretely, instead of sampling the time $t$ independently across elements of the training batch, each sequence uses a uniformly spaced time $t$. This was shown to reduce variance (in VMD, Kingma et al, and MDLM, Sahoo et al) and accelerate s

---

> ### Author Rebuttal · Authors · 2026-03-31
>
> We thank the reviewer for their thoughtful comments. We appreciate that they found the paper **”well-written, and easy to follow”**, with **”well-supported [motivation]”**. We address their concerns below.
> > **What happens in the [single]-epoch setting?**
>
>
> Due to compute constraints, we are unable to provide experimental results in a large-scale single-epoch setting. However, our results on Sudoku and TinyGSM show that **PUMA performs well even when its effective training budget is substantially less than one full epoch** (note that on our Sudoku experiments, PUMA trains on less than a full epoch anyways). Adaptive strategies, such as applying PUMA only to a high quality subset of the pretraining data while using standard MDM training on the remainder of the data, might also be helpful in the one-epoch setting.
>
> Furthermore, we have **added a new suite of fine-tuning experiments on the 7B parameter scale**, which show that **PUMA is particularly well-suited to multi-epoch instruction-finetuning**. For details on the experimental setup, we refer you to the “Experimental Setup” paragraph in our response to **knjU**. In short, on HumanEval, PUMA yields about a 10% gain within 12k iterations, whereas vanilla MDM tuning achieves at most a 3% gain. On MBPP, vanilla tuning is even less stable, while PUMA delivers a ~5% improvement.
>
> > **How can we compute an ELBO with PUMA? How does it compare to the standard MDM NELBO? How is the zero-shot likelihood?**
>
> Given PUMA’s forward process and its tied inference procedure under our policy $g_\phi$, the corresponding (N)ELBO can be written as
>
> $$\mathbb{E}\_{x_t}\mathbb{E}\_{i \sim g_\phi}\big[\mathrm{CE}(x_0^i, f_\theta^i(\cdot |x_t))\big].$$
>
> This objective, associated with $g_\phi$, reduces to the vanilla ELBO when the unmasking policy $g_\phi(x_t)$ is uniform over all masked positions.
> We would like to clarify that even an MDM trained with the vanilla forward process can be associated with alternative ELBO formulations corresponding to other choices of $g_\phi$. What matters about the forward process is whether its marginals match those induced by the inference procedure. In particular, the marginals of the vanilla forward process match random inference, while the marginals of the PUMA forward process match inference under $g_\phi$. This is an important distinction from standard continuous diffusion, where the inference process is much less flexible.
> > **Intuition for why the K-curriculum is needed? How sensitive is PUMA to [its] specific hyper-parameters?**
>
> While we could not run an ablation on the K-schedule during rebuttal, we previously observed that the training dynamics were relatively robust to differences in it. The intuition, also discussed in lines 285–290, is that early in training, when the unmasking policy is still unreliable, smaller K favors sample diversity over relying too heavily on the model-induced order.
> > **What are you doing with too-long training examples? Are you wrapping / padding the[m]?**
>
> We discard examples longer than 512 tokens. We use padding for simplicity in standard MDM training, though PUMA can also be combined with sequence packing.
>
> > **What happens with more realistic AR initialization? What is the performance of an AR baseline [...]?**
>
> We added a new experiment with an ARM initialization trained for 200k steps. PUMA reaches 44.4% final accuracy (vs. 38.1% without ARM init), while the baseline reaches 36.3% (vs. 33.6%), with a 3.7× training speedup. This suggests **PUMA especially benefits from pretrained initialization, which provides more useful training trajectories from the start**. The ARM itself is still better than PUMA with 50% final accuracy. However, this task might be particularly well-suited for ARMs, and a direct comparison to ARMs is beyond the scope of this paper.
> We provide a link to a figure visualizing our new ARM initialization experiment here: https://ibb.co/7tSpvBD3 (only one set of experiments finished before the rebuttal deadline, hence the 95% CIs in the plot are only partially available).
> > **Are you using stratified time selection?**
>
> For simplicity, we have not employed stratified time sampling yet, and have not had time to run this experiment for the initial rebuttal. However, we are happy to include this in the second response.
>
> > **How is the few / many sampling-step performance? [...] What happens with non-zero temperature sampling?**
>
> We added the following evaluation strategies:
> Confidence collapse with thresholds 0.9 and 0.95 (in each step, all tokens above the threshold are unmasked. If there are none, we unmask the two highest-confidence tokens)
> Top-5
> The accuracies (in %) are reported below (alongside our default top-2 strategy):
>
> ||top-2|conf=0.9|conf=0.95|top-5|
> |-|-|-|-|-|
> |**MDM**|33.6|32.8|32.8|33.1|
> |**PUMA**|39.1|37.7|36.9|36.5|
>
> While we haven’t had time to include an evaluation with non-zero temperature, we are happy to include it in a second rebuttal.

---

> > ### Author Rebuttal · Reviewer_n54q · 2026-04-02
> >
> > > Concern about single epoch performance
> >
> > Do you agree that, to implement PUMA **efficiently**, the training sequence $x$ must be used several times to train the MDM? This means that **in the single-epoch setting, where we only allow a single forward pass per training example $x$, PUMA cannot be implemented efficiently** (i.e. it requires two forward passes per example $x$? This is the crux of my original review. The answer should be a simple yes / no.
> >
> > > PUMA NELBO
> >
> > I am aware that several NELBO can be evaluated with a given pre-trained model (as per my original review). The question from my original review (*How does the likelihood compares after training with PUMA, if evaluated with the standard MDM NELBO*) remains un-answered. This should be relatively fast to evaluate on say 100k examples from the validation set.
> >
> > > Robustness to K
> >
> > I appreciate the answer, however I believe a deeper analysis should be included before acceptance. Regarding compute constraints, you could use shorter runs (eg up to 150k-250k steps for 5-6 schedules of K)
> >
> > > More realistic AR initialization
> >
> > From my original review, I asked about the case where >50% is spent trained as an AR model. This is more aligned with common practice (eg DiffuCoder), where the adaptation budget AR-> MDM is tiny compared to the AR pretraining budget. Hence, the new experiment does not answer my question. To minimize cost, I would be curious of what happens if you spend as little as 50k training steps with PUMA.
> >
> > > How is the few / many sampling-step performance? [...] What happens with non-zero temperature sampling?
> >
> > I believe the few-step performance is more pressing to discuss than the non-zero temperature sampling, to understand how a model trained with PUMA compares with a standard MDM across sampling budgets. The underlying question is whether PUMA helps at certain inference budgets only.

---

> > > ### Author Response · Authors · 2026-04-07
> > >
> > > We thank the reviewer for their clarifications and constructive comments. Below, we answer their remaining questions.
> > >
> > > > **Concern about single epoch performance**
> > >
> > > **Short answer: yes.** We agree that **PUMA requires more than one forward pass per sample**. However, our previous results (from our first rebuttal comment) challenge the belief that a single forward pass per sample is preferable, even in the single-epoch setting: on Sudoku, TinyGSM, and Dream-Coder-7B, we consistently observe that during the first epoch, PUMA outperforms the baseline trained on the same number of gradient steps (i.e., trained on more samples with a single forward pass per sample). This indicates that **even in the single-epoch setting, it might be beneficial to spend multiple forward passes per sample**. That being said, it remains an open question if this also applies to large-scale pretraining.
> > >
> > > > **PUMA NELBO**
> > >
> > > We evaluated the MDM NELBO on our TinyGSM validation dataset (~236k samples) with PUMA and the MDM baseline: https://ibb.co/B2YJ0xZd.
> > > As the NELBO is equal to the training objective of the baseline, it is unsurprising that the baseline attains a lower NELBO than PUMA. However, even though the NELBO does not exactly align with the PUMA training objective, PUMA is reasonably good at minimizing the NELBO as well, and attains a final NELBO of around 0.06, while the MDM baseline attains 0.05.
> > >
> > > Moreover, we conduct the same comparison on Dream-Coder-7B, with a 35k validation set and three random time steps per sequence. Here, we observe a different trend (https://ibb.co/RTbrhCfW ): **PUMA yields a lower NELBO than vanilla fine-tuning**. We hypothesize this is driven by two related factors: 1) While PUMA unmasks tokens in “easy” orders, our confidence-collapse training ensures that easy positions are unmasked early, so the model can spend more time on hard tokens; 2) random unmasking might spend compute on patterns that the model has already learned successfully. Both of these factors **can lead to more meaningful gradient updates for PUMA** and could explain the lower NELBO.
> > >
> > > > **Robustness to K**
> > >
> > > We have added additional experiments with 7 different K schedules on 5 epochs each (~225k training steps): https://ibb.co/B5PVWdk7. Each schedule starts at K=12 (except for the one that explicitly notes K=2 at the start). The plots show that while **all K schedules work reasonably well**, the one chosen for our main experimental results (blue) works best. **Increasing K too aggressively hurts performance**, suggesting a balanced schedule works best.
> > >
> > > > **More realistic AR initialization**
> > >
> > > We initialized both PUMA and a vanilla MDM from a pretrained ARM checkpoint trained for 500k steps, and then trained each model for an additional 420k steps. Therefore, roughly 54% of compute was spent on the ARM pretraining. **PUMA achieves 38.4% on TinyGSM, while the MDM achieves 30.3%**. In particular, we observe a **4.0x training speedup**.
> > >
> > > To answer the reviewer’s question about 50k training steps, the evaluation accuracy of the same **PUMA model after 50k training steps is 22.7%, while the MDM attains 15.7%**, which corresponds to a **speedup of roughly 2.3x** (i.e., PUMA attains 15.7% in 2.3x fewer steps).
> > >
> > > Overall, we note that the new results provided during the rebuttal period, including
> > > ARM initialization and 7B dLLM fine-tuning, further **strengthen PUMA’s practicality by extending our earlier experiments to more realistic scenarios** where the model is initialized with an informative unmasking policy.
> > >
> > > > **How is the few / many sampling-step performance? [...] What happens with non-zero temperature sampling?**
> > >
> > > In addition to the results in our first rebuttal, we include exhaustive evaluation results using non-zero-temperature sampling / few-step sampling, with fixed token unmasking or via confidence thresholding.
> > >
> > > |Method|PUMA|MDM Baseline|
> > > |-|-|-|
> > > |**Confidence collapse**|||
> > > |τ=0.80|36.3|31.0|
> > > |τ=0.90|37.7|32.8|
> > > |τ=0.95|36.9|32.8|
> > > |**Top-K**|||
> > > |K=2|39.1|33.6|
> > > |K=5|37.5|33.5|
> > > |K=10|28.0|22.4|
> > > |K=20|14.1|8.6|
> > > |**Temperature top-2**|||
> > > |T=0.2|37.5|33.0|
> > > |T=0.5|37.0|32.8|
> > > |T=0.7|37.8|31.8|
> > > |T=0.8|37.6|33.7|
> > > |T=1.0|39.5|33.4|
> > >
> > > “Confidence collapse” refers to the setup in which we unmask all tokens whose confidence exceeds a given threshold (or the top two tokens if none do). “Top-K” denotes the standard Top-K strategy,  corresponding to few-step sampling. “Temperature top-2” first samples tokens according to the specified temperature and then unmasks the top two tokens based on their confidence scores. **The results show that PUMA consistently outperforms the MDM baseline across all sampling configurations, suggesting that its advantage is not confined to a particular inference strategy.**
> > >
> > > We thank the reviewer again for their time. We hope that our additional results have addressed all remaining questions, and we would be grateful if **the reviewer considered raising their score if their concerns have been properly resolved**.

---

### Official Review · Reviewer_3gtn · 2026-03-10

**Soundness:** 3
**Presentation:** 4
**Significance:** 3
**Originality:** 3
**Overall Recommendation:** 4
**Confidence:** 3

**Summary:**

This paper proposes a new method, PUMA, that introduces a framework for teacher forcing to enable the masked diffusion model to use a cheaper, more realistic inference process, aiming to develop a faster masked diffusion model. By using the ground truth and experimentally showing that the unmasking process can be approximated by the policy soon, the training algorithm is stable. The paper also provides theoretical guarantees and experimental support for their idea.

**Compliance With Llm Reviewing Policy:**

Affirmed.

**Final Justification:**

The rebuttal has addressed my main concerns.

**Key Questions For Authors:**

I list below what I believe are the questions, but I would be happy to be corrected if I misunderstood any part of the work.
1. If there is some possible way to visualize or at least give a metric, I also want to see the empirical effect that the policy in the early stage is similar to the final one on TinyGSM, similar to what is shown for Sudoku.
2. Based on my understanding, the theory says the solution of mdm and mdm+puma should be equal. But all the figures show that adding puma can significantly improve accuracy. Is there any insight? Is this also a contribution or not?
3. The proof of proposition 2 last few lines I am not that understandable, I don't see how the $w_{i}(x) = \mathbb{E}[1/t\mid X]$ comes from, like I don't know where the given $X$ comes from, also I don't understand how the lemma4 works here, as new expectation is here, I hope get a more clearer proof.

Little typos, the 664 line should be the "weighting function." 676, 680 line should be $x_{t}^{\text{um}(x_{t})}$

**Limitations:**

yes

**Strengths And Weaknesses:**

## strength
1. The subject tackled is of great interest: randomly masking does need much more training process, and this is not only unused in the final inference procedure, but the misalignment may also prevent the training process from matching the inference process well.
2. The paper is well written, and I can get the main idea of the paper quite easily; the notation is easy to follow. Also, the notes and labels are clear and elegant.



## weakness
1. It's normal, but the truth is that the theory is under the ideal case, like the inference using the ground-truth unmasking posterior and choosing one position to unmask at each time, etc. But this is acceptable since the paper is primarily experimental.
2. As the limitations say, the main experiments only focus on TinyGSM. If there is more data, it will support the paper's broad claims.
3. There are a few parts of the proofs that I find difficult to understand, see question 3.
4. The experiments only run for three seeds and don't have std or other error bars. I'd love to see more seeds for stable results. At least if possible, I'd like the author to provide some error bars to give me a sense. This variance may also affect the speed-up shown in the figures.

---

> ### Author Rebuttal · Authors · 2026-03-31
>
> We thank the reviewer for the positive feedback and appreciate they found our paper to be **”well written”**, **”easy to follow”**, **”clear and elegant”**, and **”[tackling a] subject [...] of great interest”**. We address their questions below.
>
> > **I also want to see the empirical effect that the policy in the early stage is similar to the final one on TinyGSM**
>
> In addition to the corresponding results on Sudoku from the paper (compare Figures 3 and 7), we have added an analysis similar to Figure 7 for TinyGSM. For a fixed set of 100 samples and at every 40k steps during training, we save the PUMA training trajectories on these samples and compute a Kendall tau distance, normalized by the number of token-pairs, between each timestep $t$ and the final time $T=0$ (https://ibb.co/r2ZzRkRY). The plot shows that the Kendall tau distance decreases from around 0.3 at the start of training to around 0.05 very early on, and then stays at roughly 0.05 until the end of training.
>
> > **the theory says the solution of mdm and mdm+puma should be equal. But all the figures show that adding puma can significantly improve accuracy. Is there any insight?**
>
> As noted in Proposition 2, PUMA does not change the ground-truth minimizer of the training loss, as the reviewer pointed out. However, due to the non-convex nature of the training loss, **in practice, we cannot hope to converge to this true minimizer**; the empirical unmasking posterior differs from the ground-truth one. From this perspective, the higher accuracy of MDM + PUMA over vanilla MDM training can be understood as follows: **PUMA yields better predictions on the inference-relevant unmasking posterior, which acts as a beneficial inductive bias in the optimization** and leads to better empirical minimizers.
>
> > **the main experiments only focus on TinyGSM. If there is more data, it will support the paper's broad claims**:
>
> We additionally demonstrate PUMA’s efficiency on the 7B parameter scale with fine-tuning on Dream-Coder 7B. We refer to the “Experiment Setup” paragraph in our response to reviewer **knjU** for the details on this new suite of experiments. Our experiments show that on HumanEval, PUMA yields about a 10% gain within 12k iterations, whereas vanilla MDM tuning fluctuates and achieves at most a 3% gain. On MBPP, vanilla tuning is even less stable, while PUMA delivers a consistent ~5% improvement. This shows that **for large-scale LoRA finetuning, the gains of PUMA over the baseline are even more pronounced** than in the pretraining setting. The results are shown here: https://ibb.co/5xjs7sRq.
>
>
> > **if possible, I'd like the author to provide some error bars**:
>
> We added error bars across three random seeds for our experiments on both datasets and both PUMA and the baseline. The updated results are as follows (with 95% CIs):
>
> | | Sudoku | TinyGSM |
> |--------------|------------------------------|------------------------------|
> | PUMA | 39.1 $\pm$ 1.0 | 93.4 $\pm$ 0.6 |
> | Vanilla MDM | 33.6 $\pm$ 1.2 | 87.1 $\pm$ 2.6 |
> | Speedup | 2.3$\times$ (prev.: 2.5$\times$) | 1.7$\times$ (prev.: 1.4$\times$) |
>
> We will also add error bars to our plots in the updated version of the paper, including the results for ARM initialization. We provide a link to the updated Figure 1 from the paper here: https://ibb.co/tMnTDQkh.
>
>
> > **The proof of proposition 2 last few lines [...] I don’t understand**
>
>
> We thank the reviewer for pointing this out, as the proof contains some notational inaccuracies. We provide clarifications below.
> Let $X=x_t$ and $Y=x_0^i$ for a fixed coordinate $i$. Then the $i$-th coordinate’s contribution to the loss is
> $$L_i(\theta) = \mathbb{E}\_{X,t,Y} [ \frac{1}{t}  \mathbf{1}[X^i = m]  (- \log f_\theta^i(Y | X)) ].$$  Earlier in the proof, we saw that by Bayes’ rule,
> $$\mathbb{P}(x_0 \mid x_t,t) \propto \alpha_t(x_t)\,\mathbf{1}[x_0^{\mathbf{um}(x_t)} = x_t^{\mathbf{um}(x_t)}] p_{\mathrm{data}}(x_0)$$ and since $\alpha_t(x_t)$ does not depend on $x_0$, and the right hand side does not depend on $t$, this implies $P(Y=\cdot \mid X,t)=P(Y=\cdot \mid X)$. Therefore, we can rewrite
> $$L_i(\theta)=\mathbb{E}\_{X,Y} [\mathbb{E}\_t [\frac{1}{t}\mathbf{1}[X^i=m]\mid X] (-\log f_\theta^i(Y\mid X))]
> =\mathbb{E}\_{X,Y}[w_i(X) (-\log f_\theta^i(Y\mid X))]$$
>  where $w_i(X) := \mathbb{E}_t\left[\frac{1}{t} \mathbf{1}[X^i=m] \mid X\right] $. Since this is a nonnegative function of $X$, we can apply Lemma 4 to $L_i(\theta)$, which finishes the proof.
>
> At a high level, the proof works by separating the argument into two simple steps. We first show that the modified forward process does not change the posterior target conditioned on a partially revealed sequence, since the additional factor in Eq. (2) is independent of $x_0$ and cancels out under Bayes’ rule; we then observe that, coordinate-wise, the loss is just a weighted cross-entropy with respect to that same posterior, so Lemma 4 gives the desired minimizer immediately.

---

> > ### Author Rebuttal · Reviewer_3gtn · 2026-04-03
> >
> > I thank the authors for their effort in addressing these questions. My concerns have been adequately addressed, and I maintain my positive recommendation.

---

> > > ### Author Response · Authors · 2026-04-07
> > >
> > > We thank the reviewer for the overall positive evaluation again and for acknowledging our rebuttal.

---

### Official Review · Reviewer_knjU · 2026-03-13

**Soundness:** 2
**Presentation:** 2
**Significance:** 3
**Originality:** 3
**Overall Recommendation:** 4
**Confidence:** 4

**Summary:**

This paper proposed PUMA, a specialized training strategy that accelerates the training of masked diffusion models (MDMs). Specifically, PUMA targets the mismatch between training and inference of MDMs and redesigns the forward masking process to better align with the inference of MDMs. By leveraging an unmasking policy, which is just the MDM itself, PUMA decides the masked positions based on the scores given by the policy and masks positions with lower scores. By measuring the test accuracy on Sukodu and GSM8K, PUMA accelerates the training by up to $2.5\times$ to get the same performance.

**Compliance With Llm Reviewing Policy:**

Affirmed.

**Key Questions For Authors:**

1. Although I understand the mindset and the proposed theoretical guarantee of PUMA on training-inference alignment, my concern is the robustness at the beginning of MDM training. Specifically, PUMA utilizes the logits of MDM itself as a metric for masking. However, at the beginning of MDM training, the MDM does not obtain a high capability of denoising, and the logits can be inaccurate and highly fluctuating, which is not likely to be a good indicator for masking. How does the MDM training prevent collapsing and converging to a suboptimal? Could the authors provide a deep explanation?

1. Is the claim of "using the current model for teacher-forced chains is a reasonable proxy already early in training" rigorous?

1. According to the paper, the number of unmasked tokens $|S|$ should roughly be $L/K$. However, in the experiments, $K$ is set from 12 to 42, and $|S|$ is set to 2 or 3, while $L=512$. Could the authors clarify this part?

1. In Figure 4, for different inference unmasking policies, including Top-K margin and Entropy, are they also applied during the training stage? Or is the probability confidence score used for unmasking during training?

1. The metric used for training acceleration measurement is not solid enough to me. This paper only uses the GSM8K test accuracy for measurement, and the training dataset is also from the same domain and converted to the same format. Under random masking, MDMs can learn to denoise multiple permutations of masked sequences. For one specific task, decoding might follow some specific orders, and only learning those orders during training can preserve a similar reasoning capability on that task while using fewer training iterations. However, the trained MDM might not obtain a strong generalizability to other tasks where other decoding orders are preferred for better reasoning. Thus, it is unclear if PUMA can accelerate the training of a generalizable MDM that shows strong performance on a vast spectrum of tasks. My concern is not about the training set diversity and scale, considering the feasibility of pre-training, but the measurement of training quality. Do the authors also have the test accuracy on other benchmarks during training? And could the authors provide a further discussion on this point?

**Limitations:**

1. The proposed method is only applied to a 125M MDM. It is unclear if PUMA can also accelerate the training of larger MDMs. It is necessary to include a training experiment (at least post-training, considering the unaffordable cost of pre-training) applied to larger models like LLaDA and Dream. In my opinion, PUMA is more likely to contribute to the post-training efficiency, as the pre-trained MDMs have already obtained decent denoising ability and can provide more accurate signals for masking.

1. The experiments are not extensive enough. Only the test accuracy on one task, which is also from the same domain as the training dataset, is used to measure if PUMA achieves the same training quality with fewer training iterations.

**Strengths And Weaknesses:**

### Strengths
1. Overall, the proposed method is intuitive and reasonable.
1. The motivation of this paper is clear. The mismatch between training and inference of MDMs is an important challenge in the research community.
1. In addition to empirical experimental results, a theoretical analysis is also provided to support the design of the proposed method.


### Weaknesses
1. The whole structure and writing of this paper are not clear enough, including
    - The presentation of the experimental results is disorganized. The description of the experiments in the main text lacks clarity, and the references to the figures and tables are scattered and difficult to follow.
    - For example, multiple figures are referenced in one paragraph, but they do not show a strong connection, and there is no clear description that explains why these results support the authors' claim.
    - The excessive use of multi-colored highlights throughout the manuscript creates visual clutter, which significantly detracts from the readability of the paper, ranther than clarifying the points.

1. The experiments are not extensive enough. It does not support the claim that PUMA goes beyond task-specific performance gains and improves general training-time efficiency. Please see **Questions**.

1. Besides, PUMA is only applied to a 125M MDM. It is unclear if PUMA is a scalable training strategy.

---

> ### Author Rebuttal · Authors · 2026-03-31
>
> We are grateful to the reviewer for their valuable feedback on our paper. We appreciate that they found our method **”intuitive and reasonable”**  and addressed an **”important challenge in the research community”**. At the same time, the reviewer raises valid concerns, which we address below.
>
>  ### **Concern 1: Scalability and generality of the empirical results**
>
> > **The proposed method is only applied to a 125M MDM. It is unclear if PUMA can also accelerate the training of larger MDMs. [...] [I]t is unclear if PUMA can accelerate the training of a generalizable MDM that shows strong performance on a vast spectrum of tasks.**
>
> We **address both of these concerns with a post-training experiment on Dream-Coder-Base-7B.**
>
> **Experiment Setup.** We collect 700K training pairs from opc-sft-stage2-educational, KodCode-V1-SFT-4o, and OpenCodeInstruct, with all sequences padded to a maximum length of 640. We then fine-tune Dream-Coder-Base-7B with a frozen backbone and LoRA rank 128 (~300M trainable parameters). We compare vanilla training against PUMA, with a fixed K-schedule of K=42 and the confidence thresholding (0.9). All runs are conducted on 8 H100 GPUs with a global batch size of 256. Following the exact evaluation protocol of the Dream-Coder paper, we measure HumanEval and MBPP accuracy every 2,000 training steps.
>
> **Results.** The results are shown here: https://ibb.co/5xjs7sRq. On HumanEval, PUMA yields about a 10% gain within 12k iterations, whereas vanilla MDM tuning fluctuates and achieves at most a 3% gain. On MBPP, vanilla tuning is even less stable, while PUMA delivers a consistent ~5% improvement. This shows that **for large-scale LoRA finetuning, the gains of PUMA over the baseline are even more pronounced** than in the pretraining setting. We attribute this to the fact that LoRA provides a smaller parameter space to optimize, which makes it even more important to provide a rich training signal, which PUMA achieves by limiting training to inference-aligned predictions.
>
> These experiments address both of the reviewer’s concerns: We show that **PUMA is scalable to 7B models**, and that it **generalizes to broader tasks**, such as the HumanEval and MBPP benchmarks, while being substantially better than fine-tuning with vanilla MDM loss.
>
>
> ### **Concern 2: Unmasking policy at the early stage**
>
> > **[A]t the beginning of MDM training, [...] the logits can be inaccurate [...] which is not likely to be a good indicator for masking. How does the MDM training prevent collapsing [...]?**
>
> **Our K-schedule acts as an empirical mitigation strategy for inaccurate logits early in training** (lines 285–290 in the paper): by starting with smaller values of K (i.e., unmasking more tokens at a time), we rely less on the model’s unmasking order early on, while exploiting what the model has learned at later stages by increasing K.
>
> While there is **no theoretical guarantee that PUMA training will not collapse to harmful unmasking orders, we see strong empirical evidence that this does not happen**. This is likely because the training trajectories depend purely on the ranking of token confidences predicted by the model, which can align with the inference ranking earlier than the logits themselves (compare lines 196–212 in the paper). We already provided empirical evidence for this on Sudoku in Section 4.1. To further strengthen this point, we conduct a similar analysis on TinyGSM. For a fixed set of 100 samples and at every 40k steps during training, we save the PUMA training trajectories on these 100 samples and compute a Kendall tau distance (that allows for ties), normalized by the number of token-pairs, between each training timestep and the final timestep. We provide a link to the plot here: https://ibb.co/r2ZzRkRY. The plot shows that the Kendall tau distance decreases from around 0.3 at the start of training to around 0.05 very early on, and then stays at roughly 0.05 until the end of training.
>
> ### **Other Questions**
>
> > **The number of unmasked tokens $|S|$ should roughly be $L/K$. However, in the experiments, $K$ is set from 12 to 42, and $|S|$ is set to 2 or 3 [...] Could the authors clarify this part? [...] [Are] top-K margin and Entropy [...] also applied during the training stage?**
>
> At training, we increase $K$ from 12 to 42 (on TinyGSM), and unmask $|S|=L/K$ tokens at each step, in addition to all tokens above the 0.9 confidence threshold; at inference, we unmask $|S|=2,3$ tokens at a time. In other words, the precise unmasking policies used for training and inference could differ, including $|S|$ and how we choose $S$, e.g. via top-K margin or entropy. We also refer the reviewer to our response to reviewer **n54q**, where we evaluate additional sampling strategies at inference, including confidence collapse for thresholds 0.9 and 0.95.
>
> We would also like to acknowledge the remarks on writing clarity, in particular in our experiments section, and will improve the presentation in a revised version!

---

> > ### Author Rebuttal · Reviewer_knjU · 2026-04-01
> >
> > Thank the authors for the responses. Overall, my primary concerns are about the scalability and the stability at the early training stage. For scalability, the authors provide a post-training experiment on a larger model, which shows good performance. For stability, the authors provide an additional trajectory distance study, which further supports the stability from an empirical perspective. With those concerns being resolved, I think this paper is technically solid, and I am glad to raise my score to 4. Besides, I am looking forward to a theoretical guarantee that PUMA training will not collapse in the future.

---

> > > ### Author Response · Authors · 2026-04-07
> > >
> > > We thank the reviewer for acknowledging our rebuttal on PUMA's scalability and stability analysis. We're happy to include the points discussed in the review and our rebuttal in the revised version of our paper.

---

### Decision · Program_Chairs · 2026-04-30

**Decision:**

Accept (regular)

**Comment:**

The paper proposes a modification of the masking process in masked diffusion models, with the goal of better aligning training and inference. This leads to faster training and, in practice, improved efficiency.

All reviewers are overall satisfied with the paper. They agree the problem is relevant and that the approach is reasonable and technically sound. The empirical results support the claims, and the method is simple enough to be useful. After the rebuttal, most concerns were addressed to a good extent, and reviewers kept their positive stance, all leaning toward weak acceptance.

That said, some points are still not fully resolved. In particular, there are open questions about scalability to larger models, how sensitive the method is to the choice of K, the reliance on single-epoch settings, and the role of the threshold selection. There are also some issues with clarity in the writing. The rebuttal helped, but these aspects remain somewhat unclear.